# High-Dimensional Online Change Point Detection with Adaptive Thresholding and Interpretability

## Abstract

Change point detection (CPD) identifies abrupt and significant changes in sequential data, with applications in human activity recognition, financial markets, cybersecurity, manufacturing, and autonomous systems. Traditional CPD methods often face computational challenges in high-dimensional settings and typically provide limited explanations for detected changes, which can restrict their practical usability. This paper introduces a CPD framework that improves scalability and interpretability by leveraging the Sliced Wasserstein (SW) distance. Our contributions are fourfold: (1) we transform multivariate sequential data into one-dimensional scores using the SW distance, making the resulting representation compatible with existing CPD methods; (2) we analyze the distributional behavior of random slices of the SW distance and show that, under suitable assumptions, they can be approximated by a Gamma distribution, providing a principled basis for threshold calibration; (3) we propose a self-adapting online CPD algorithm that combines this SW-based score with an adaptive quantile-based threshold; (4) we introduce a model-specific framework for generating contrastive explanations for annotated change points. Empirically, our method reduces false positives by at least 48% on average compared with popular online and offline CPD baselines, while maintaining competitive or superior detection performance[1]. At the same time, it produces interpretable change-point annotations, making it practical for deployment in high-stakes applications.

## 1 Introduction

Change point detection (CPD) is a fundamental problem in statistical analysis, focusing on identifying abrupt and significant changes in the underlying data-generating processes of sequential data. These changes can signal shifts in critical properties, such as distributions, relationships, or trends, making CPD pivotal in fields where timely detection of such shifts is crucial. Closely related to concept drift detection (Gama et al., 2014; Harel et al., 2014; Lu et al., 2018), CPD encompasses scenarios of both abrupt and gradual changes, with a direct impact on the accuracy and reliability of machine learning models and deployed systems. However, existing CPD methods are insufficient in both scaling to high dimensions and providing meaningful explanations, which poses a significant gap addressed by our approach.

The significance of CPD becomes evident in its multitude of real-world applications. In *human activity recognition*, it can identify transitions between states, such as detecting when a person moves from walking to running (Xia et al., 2020). In *financial markets*, CPD is essential for spotting regime shifts, such as the transition from a bull to a bear market, enabling traders and algorithms to adjust strategies (Kim et al., 2022; Carvalho & Lopes, 2007; Chen & Gupta, 1997; Nystrup et al., 2016). In *cybersecurity*, CPD helps detect anomalies, such as cyberattacks or data breaches, by identifying abrupt deviations in network traffic (Kurt et al., 2018; Polunchenko et al., 2012). Similarly, in *manufacturing quality control*, CPD can pinpoint defects or process anomalies to minimize waste and downtime. Furthermore, in *autonomous driving*, detecting changes in environmental conditions or sensor data ensures safe operation under dynamic conditions (Ferguson et al., 2014; Galceran et al., 2017). These examples underscore the critical role of CPD in enhancing decision-making and ensuring the safety, efficiency, and reliability of systems across domains.

---

[1]Code is available at `https://anonymous.4open.science/r/SWCPD-7022`.

Despite its utility, CPD faces significant challenges when applied to high-dimensional data, where both scalability and explainability are becoming increasingly challenging. Traditional methods often rely on comparing probability distributions or distances between data segments to detect changes (Aminikhanghahi & Cook, 2017; Lu et al., 2018). While effective in lower-dimensional settings, these methods struggle with computational efficiency and scalability in higher-dimensional spaces. For instance, the exact computation of the Wasserstein distance for multivariate data scales as $\mathcal{O}(n^3 \log(n))$, making it impractical for large datasets. Similarly, the computation of $U$- and $V$-statistics for the Maximum Mean Discrepancy (MMD) also scales quadratically in time. Alongside the computational aspects, most CPD methods fail to provide interpretable change points, narrowing down the root cause of the drifts.

To address the lack of interpretable change point detection tailored for high-dimensional data, the Sliced Wasserstein (SW) distance (Bonneel et al., 2015) offers a promising alternative. Instead of computing a high-dimensional optimal transport directly, we can repeatedly project onto a single dimension, where Wasserstein distance has a closed form, and then average the results. By leveraging the closed-form expression of the Wasserstein distance for one-dimensional distributions, the SW distance reduces the computational complexity to $\mathcal{O}(n \log(n))$ by averaging over the Wasserstein distances of random one-dimensional projections. Additionally, by leveraging the geometric properties of the random projections, we can provide contrastive explanations for detected change points.

In this work, we bridge this gap by introducing a self-adapting online CPD algorithm that combines the SW distance with an adaptive quantile-based threshold. Our contributions are as follows:

1. **A Self-Adapting Online CPD Algorithm with Adaptive Thresholding (§4).** We propose a self-adapting online CPD algorithm that dynamically adjusts its threshold based on a quantile-based threshold. This enables robust and adaptive detection of change points in streaming high-dimensional data without requiring a fixed global detection threshold.

2. **Distributional Analysis of SW-Based Random Slices (§3).** We analyze the distributional behavior of random slices derived from the SW distance and show that, under suitable assumptions, they can be approximated by a Gamma distribution. This provides a principled motivation for threshold calibration and helps explain the empirical behavior of the proposed detection statistic.

3. **Contrastive Explanations for Change Points Using Geometric Properties of SW Distance (§4.1).** We develop a novel, model-specific framework for generating contrastive explanations of detected change points. By leveraging the geometric properties of random projections, we provide fine-grained insights into which features contribute most to distributional shifts, enhancing interpretability.

4. **Competitive Performance with Interpretability (§5.2)** We evaluate our proposed framework on multiple real-world datasets and show that it achieves competitive or superior performance compared to leading online and offline CPD methods. In particular, it reduces false positives while also producing interpretable change-point annotations, supporting its practical use in high-dimensional applications.

## 2 Related Work

**Online change point detection.** Change point detection can be grouped into parametric and nonparametric methods (Truong et al., 2020). Parametric methods assume that the data is drawn from some parametric family of probability distributions. Nonparametric approaches do not impose distributional assumptions. One of the most prominently known parametric approaches is the cumulative sum (CUSUM) method (Page, 1954). Over the last years, several extensions of CUSUM were introduced (Alippi & Roveri, 2006; Romano et al., 2023; Yu et al., 2023). Another popular parametric branch of change point detection are Bayesian methods including (Fearnhead & Liu, 2007; Knoblauch et al., 2018). Nonparametric methods are often based on test statistics derived by distances, including Euclidean distances (Matteson & James, 2014; Madrid Padilla et al., 2019) or divergence measures e.g. MMD (Gretton et al., 2012; Harchaoui et al., 2013; Li et al., 2019) or test-statistics based on density-ratio estimation (Sugiyama et al., 2008; Kanamori

et al., 2009; Yamada et al., 2013; Liu et al., 2013b). More recently, deep generative models (Chang et al., 2019; De Ryck et al., 2021) and density-ratio estimation based on deep neuronal networks (Hushchyn et al., 2020; Hushchyn & Ustyuzhanin, 2021) were also used for sequential change point detection.

**Optimal transport based change detection.** Over the past few years, optimal transport has become a popular choice for comparing two distributions. Naturally, optimal transport-based metrics, such as the Wasserstein distance or Sliced Wasserstein distance, can also be applied for sequential change point detection. This includes Cheng et al. (2020a), which proposes a change point detection framework computing the Wasserstein distance between a sliding window relying on a fixed threshold to detect changes. Similar approaches were introduced in Faber et al. (2021; 2022). In Cheng et al. (2020b), this framework was refined using a matched filter test statistic. Furthermore, one of the proposed test statistics is the Sliced Wasserstein distance, which is combined with a fixed threshold. Our work differs by introducing an adaptive threshold and primarily investigating the Sliced Wasserstein distance as a tool for interpretability.

**Interpretability through random projections.** The motivation behind utilizing random projection is the lower computational cost for the Wasserstein distance. In Wang et al. (2021), a projected Wasserstein distance was introduced, which finds a k-dimensional subspace through linear projections and calculates the Wasserstein distance in the lower-dimensional space. Analogously, in Wang et al. (2022), the kernel projected Wasserstein distance was motivated as a non-linear alternative to Wang et al. (2021). Both approaches reduce the computational complexity and facilitate interpretability in a two-sample test. Our proposed framework goes beyond a single iteration to find a specific projection direction, maximizing the Wasserstein distance between projected samples. We propose an iterative approach to identify the most discriminative feature, leading to a more comprehensive and detailed explanation of the underlying drift. Recent literature (Hinder & Hammer, 2023) has highlighted that random projections are not universally beneficial for drift detection. At the same time, several studies support their use in high-dimensional two-sample testing (Rabanser et al., 2019; Wang et al., 2021). Taken together, these findings suggest that random projections should be viewed as a computational–statistical trade-off rather than as a transformation that uniformly improves detection performance.

## 3 Problem Setup

The general problem of CPD involves determining abrupt changes in a time series. We denote the time series $\mathcal{D} = \{x_t \in \mathbb{R}^d : t \in [T]\}$ with $[T] = \{1, 2, \ldots, T\}$ and assume that the time series follows some unknown underlying distribution $P$. The goal is to identify all timestamps $t_* \in [T]$ where the underlying distribution changes from $P$ to $Q$, such that $t \leq t_* : x_t \sim P$ and $t > t_* : x_t \sim Q$. Consider $P, Q$ to be two probability distributions with $p$ finite moments. The Wasserstein distance, denoted as, $W_p^p(\mathbb{P}, \mathbb{Q})$ has a closed expression for univariate distributions,

$$W_p^p(P, Q) = \int_0^1 |F^{-1}(u) - G^{-1}(u)|^p \mathrm{d}u$$

where $F^{-1}, G^{-1}$ are the inverse CDF of $P$ and $Q$ respectively. The sliced Wasserstein distance (SW) exploits this closed expression by averaging over the Wasserstein distance between infinitely many random one-dimensional projections of $P$ and $Q$. In particular, for any direction $\theta \in \mathbb{S}^{d-1}$, we define the projection of $x \in \mathbb{R}^d$ as $T^\theta(x) = \langle x, \theta \rangle$ and denote the projected distribution with $P_\theta = T_\#^\theta P$, where $\#$ is the push-forward operator, defined as $T_\# P(A) = P(T^{-1}(A))$ for any Borel set $A \in \mathbb{R}^d$. Let us denote $\lambda$ the uniform measure on $\mathbb{S}^{d-1} = \{\theta \in \mathbb{R}^d : ||\theta||_2 = 1\}$, then the $p$ Sliced Wasserstein distance between $P$ and $Q$ is defined as

$$SW_p^p(P, Q) = \int_{\mathbb{S}^{d-1}} W_p^p(P_\theta, Q_\theta) \mathrm{d}\lambda(\theta). \tag{1}$$

In practice, the computation of the SW boils down to a Monte Carlo approximation by uniformly sampling projection parameters $\{\theta_\ell\}_{\ell=1}^L$ on $\mathbb{S}^{d-1}$ and average over the one-dimensional Wasserstein distances obtained. The accuracy of this estimator heavily relies on the variance of the projected Wasserstein distance (Nietert et al., 2022).

Based on the following result, we derive the adaptive threshold calibration, which is based on the MoM estimated parameters of a Gamma distribution.

Let $X \sim P$ and $Y \sim Q$ be random vectors in $\mathbb{R}^d$ with means $\mu_P, \mu_Q \in \mathbb{R}^d$ and covariance matrices $\Sigma_P, \Sigma_Q \in \mathbb{R}^{d \times d}$ (symmetric p.s.d.). Let $\theta \sim \mathrm{Unif}(\mathbb{S}^{d-1})$ be independent of $(X, Y)$. Denote,

$$S_d(\theta) := W_2^2(P_\theta, Q_\theta).$$

In the following, $S_d(\theta_\ell)$ is evaluated for i.i.d. $\theta_\ell$ and we model the empirical slice set $\{S_d(\theta_\ell)\}_{\ell=1}^L$ by a Gamma law.

We assume the following high-dimensional regime.

(A1) (Moments) $X$ and $Y$ have finite third moments, i.e. $\mathbb{E}\|X\|_2^3 < \infty$ and $\mathbb{E}\|Y\|_2^3 < \infty$.

(A2) (Spherical CLT for projections) As $d \to \infty$, the one-dimensional projections satisfy a Gaussian approximation in the following sense: conditionally on $\theta$, the laws of $\langle X, \theta \rangle$ and $\langle Y, \theta \rangle$ are asymptotically close (e.g. in Kolmogorov distance) to $\mathcal{N}(m_P(\theta), v_P(\theta))$ and $\mathcal{N}(m_Q(\theta), v_Q(\theta))$ with

$$m_P(\theta) = \theta^\top \mu_P, \quad v_P(\theta) = \theta^\top \Sigma_P \theta,$$
$$m_Q(\theta) = \theta^\top \mu_Q, \quad v_Q(\theta) = \theta^\top \Sigma_Q \theta,$$

(This holds exactly if $P, Q$ are Gaussian; it also holds under standard high-dimensional projection CLTs for many non-Gaussian families.)

(A3) (Spectral regularity) There is a constant $C$ independent of $d$ such that $\|\Sigma_P\|_{\mathrm{op}} \leq C$, $\|\Sigma_Q\|_{\mathrm{op}} \leq C$ and

$$\frac{1}{d}\mathrm{tr}(\Sigma_P) \to \tau_P, \quad \frac{1}{d}\mathrm{tr}(\Sigma_Q) \to \tau_Q,$$
$$\frac{1}{d}\mathrm{tr}(\Sigma_P^2) \to \kappa_P, \quad \frac{1}{d}\mathrm{tr}(\Sigma_Q^2) \to \kappa_Q,$$

for some finite $\tau_P, \tau_Q, \kappa_P, \kappa_Q > 0$.

(A4) (Asymptotic decorrelation) The pair of quadratic forms $v_P(\theta) = \theta^\top \Sigma_P \theta$ and $v_Q(\theta) = \theta^\top \Sigma_Q \theta$ is asymptotically jointly normal after centering and scaling, with a limiting covariance that is $O(1)$; and the linear form $\theta^\top(\mu_P - \mu_Q)$ is asymptotically independent of $(v_P(\theta), v_Q(\theta))$. (These properties hold, for instance, when $\theta = g/\|g\|$ with $g \sim \mathcal{N}(0, I_d)$ and $\mu_P - \mu_Q$ is orthogonal in the eigenbasis in which $\Sigma_P, \Sigma_Q$ are simultaneously diagonalizable, or more generally under standard isotropic random direction asymptotics.)

**Theorem 3.1.** *(Asymptotic law of $S_d(\theta)$) Assume (A1)–(A4). Let $\delta := \mu_P - \mu_Q$. Define the random vector*

$$U_d(\theta) := \begin{pmatrix} u_{1,d}(\theta) \\ u_{2,d}(\theta) \end{pmatrix} := \begin{pmatrix} \theta^\top \delta \\ \sqrt{\theta^\top \Sigma_P \theta} - \sqrt{\theta^\top \Sigma_Q \theta} \end{pmatrix},$$

*and let $\sqrt{d}\,\Delta_d \to m_2$, where $\Delta_d = \sqrt{\frac{1}{d}\mathrm{tr}(\Sigma_P)} - \sqrt{\frac{1}{d}\mathrm{tr}(\Sigma_Q)}$, with $m_2 \in \mathbb{R}$. Then, under (A2), the population slice statistic satisfies*

$$S_d(\theta) = W_2^2(P_\theta, Q_\theta) = \left(u_{1,d}(\theta)\right)^2 + \left(u_{2,d}(\theta)\right)^2 + r_d(\theta), \tag{2}$$

*where $r_d(\theta) \to 0$ in probability as $d \to \infty$ (the error stems only from the projection-to-Gaussian approximation in (A2)). Moreover, as $d \to \infty$, there exist centering/scaling constants such that*

$$\begin{pmatrix} \sqrt{d}\,u_{1,d}(\theta) \\ \sqrt{d}\,u_{2,d}(\theta) \end{pmatrix} \Rightarrow \mathcal{N}\left(\begin{pmatrix} 0 \\ m_2 \end{pmatrix}, \Omega\right),$$

*some $2 \times 2$ covariance matrix $\Omega \succeq 0$. Consequently, $d\,S_d(\theta)$ converges in distribution to a (possibly noncentral) generalized chi-square random variable, i.e.*

$$d\,S_d(\theta) \Rightarrow Z^\top A Z,$$

*where $Z \sim \mathcal{N}(\mu_Z, \Sigma_Z)$ is Gaussian and $A \succeq 0$ is a fixed matrix. In particular, the limiting law is supported on $\mathbb{R}_+$.*

The local variance-shift condition ($\sqrt{d}\Delta_d \to m_2$) is necessary because, if the average projected variances differ by a fixed nonzero amount, then $\sqrt{d}\,u_{2,d}(\theta)$ generally does not have a finite limiting mean and the scaled statistic $dS_d(\theta)$ may diverge. The condition is satisfied under equal average projected variances, in particular for pure mean-shift drifts with $\Sigma_P = \Sigma_Q$ and more generally when the difference in average projected standard deviations is of order $\mathcal{O}(d^{-1/2})$. In the following, we model random slices $\{S_d(\theta_\ell)\}_{\ell=1}^L$ by a Gamma distribution. Theorem 3.1 shows the correct limit is generalized chi-square in general. It becomes *asymptotically Gamma* for mean-shift regimes, which is also the regime where random projections are most diagnostic for change points.

**Corollary 3.2** (Mean-shift dominated drift $\Rightarrow$ Gamma limit). *Assume the conditions of Theorem 3.1 and, in addition,*

$$\sqrt{\theta^\top \Sigma_P \theta} - \sqrt{\theta^\top \Sigma_Q \theta} = o_p(d^{-1/2}) \quad as\ d \to \infty,$$

*(i.e. the variance term is negligible compared to the mean term; this holds for mean-shift drifts with nearly unchanged second moments). Then*

$$d\,S_d(\theta) = d\,(\theta^\top \delta)^2 + o_p(1) \ \Rightarrow\ \|\delta\|_2^2\,\chi_1^2.$$

*Equivalently, the limit is:*

$$d\,S_d(\theta) \ \Rightarrow\ \Gamma\left(\frac{1}{2},\ \frac{1}{2\|\delta\|_2^2}\right).$$

*where $\Gamma(\alpha, \beta)$ denotes a Gamma distributions with shape- and rate parameter $\alpha, \beta$.*

*Proof.* Under the stated condition, $u_{2,d}(\theta) = o_p(d^{-1/2})$, hence $S_d(\theta) = (\theta^\top \delta)^2 + o_p(d^{-1})$ by equation 2. By the spherical CLT (Step 2 in the proof of Theorem 3.1), $\sqrt{d}\,\theta^\top \delta \Rightarrow \mathcal{N}(0, \|\delta\|_2^2)$. Squaring yields $d(\theta^\top \delta)^2 \Rightarrow \|\delta\|_2^2 \chi_1^2$. The Gamma statement follows from the identity $\chi_1^2 \sim \Gamma(1/2, 1/2)$. $\square$

Even when the full limit is a generalized chi-square (Theorem 3.1), the slice statistic is nonnegative and often well-approximated by a Gamma distribution in practice. This is precisely the modeling assumption used by SWCPD: given i.i.d. slice samples $\{S_d(\theta_\ell)\}_{\ell=1}^L$, fit a Gamma distribution by matching the empirical mean and variance using the Method of Moments (MoM),

$$\widehat{\alpha} = \frac{\overline{S}^2}{\mathrm{Var}(S)}, \qquad \widehat{\beta} = \frac{\overline{S}}{\mathrm{Var}(S)}, \tag{3}$$

## 4 Proposed Detection Method

In the following, we describe an adaptive online change point detection method (SWCPD) that monitors the cumulative Sliced Wasserstein distances against a dynamic quantile-based threshold. Algorithmically, SWCPD follows a CUSUM-style monitoring principle applied to SW-induced slice statistics. The novelty of the method lies not in the use of cumulative monitoring itself, but in the Gamma-motivated adaptive calibration of the threshold grounded by the random slices of the SW distance, and the accompanying contrastive explanation mechanism.

At each time step, we fit a Gamma distribution to the collection of projected Wasserstein distances $\{S_d(\theta_\ell)\}_\ell^L$ and compute an adaptive control limit $\kappa_t(1 - q)$, defined as the $(1 - q)$-quantile of the fitted Gamma distribution, where $q$ controls the nominal upper-tail probability under the fitted model.

(1) UPDATE CUMULATIVE SUM: We compute the expected value of the test statistic as follows

$$C_t = C_{t-1} + \frac{\widehat{\alpha}}{\widehat{\beta}}.$$

(2) PROPAGATE MoM ESTIMATES: In a sliding window, there are dependencies between successive data windows. We smooth past MoM estimates using a moving average over the most recent $m = \min\{K_{max}, t\}$

steps with

$$\mathbb{E}[\hat{\alpha}_{t+1}|C_t] = \frac{1}{m}\sum_{i=t-m}^{t}\hat{\alpha}_i \quad \mathbb{E}[\hat{\beta}_{t+1}|C_t] = \frac{1}{m}\sum_{i=t-m}^{t}\hat{\beta}_i.$$

(3) BOUND CUMULATIVE SUM: We use the smoothed MoM estimates to bound the next step in the cumulative sum via the quantile of the derived Gamma distribution:

$$\mathbb{E}[C_{t+1}|C_t] = C_t + \mathbb{E}\left[\frac{\hat{\alpha}_{t+1}}{\hat{\beta}_{t+1}}\Big|C_t\right] \leq C_t + \kappa_{t+1}(1-q)$$

where $\kappa_{t+1}(1-q)$ denotes the $(1-q)$-quantile of $\Gamma(\hat{\alpha}_{t+1}, \hat{\beta}_{t+1})$.

(4) VALIDATE DEVIATIONS: After observing a new sample, we update $C_{t+1}$, and compare it against the upper bound. If it exceeds the bound, a change point is detected. The MoM estimates are then updated using the new data. Under stable dynamics and a well-calibrated fitted model, the next-step Sliced Wasserstein statistic exceeds $\kappa_{t+1}(1-q)$ with probability approximately $q$. This yields an adaptive quantile thresholding mechanism.

The resulting threshold should be interpreted as an adaptive calibration rule rather than a finite-sample hypothesis test with guaranteed type-I error control. In particular, the theoretical result concerns population slice statistics in a high-dimensional asymptotic regime, whereas the online detector operates with empirical windows, finitely many projections, and overlapping observations. These factors can affect calibration in finite samples. For a detailed overview, we outline the proposed detection procedure in Algorithm 1. At each time step $t$, we partition the sliding window into two non-overlapping consecutive subsets (or sub-windows). Specifically, for window length $w$ at $t$, we set $X_{ref} = \{x_t, \ldots, x_{t+\lfloor w/2 \rfloor -1}\}$ and $X_{cur} = \{x_{t+\lfloor w/2 \rfloor}, \ldots, x_{t+w}\}$. Both windows induce empirical distributions $\widehat{P}, \widehat{Q}$ respectively. In practice, the computation of the SW distance boils down to a Monte Carlo approximation by uniformly sampling projection parameters $\{\theta_l\}_{l=1}^L$ on $\mathbb{S}^{d-1}$ and averaging over the one-dimensional Wasserstein distances $L^{-1}\sum_{l=1}^L W_2^2(\widehat{P}_\theta, \widehat{Q}_\theta)$. As noted, we write $S_d(\theta) = W_2^2(\widehat{P}_\theta, \widehat{Q}_\theta)$ and collect the one-dimensional slices (line 8) $S_t = \{S_d(\theta_l)\}_{l=1}^L$, we show that under given assumptions $S_t$ has a Gamma limit. This means that $\mathrm{SWD}_t = \mathbb{E}[S_t] = \alpha_t/\beta_t$ (line 9), thus the MoM estimator (line 22) reads $\hat{\alpha}_t = \mathbb{E}[S_t]^2/\mathrm{Var}(S_t)$ and $\hat{\beta}_t = \mathbb{E}[S_t]/\mathrm{Var}(S_t)$.

## 4.1 Interpretability

We interpret $S_d(\theta_\ell)$ as the loss associated with projection direction $\theta_\ell$, where the loss quantifies the Wasserstein distance between the corresponding one-dimensional projections. This establishes a direct link between projection directions and distributional discrepancy. We use this link to derive a feature-importance score by averaging the absolute projection parameters corresponding to the slices above the $q$-quantile of $S_L = \{S_d(\theta_\ell)\}_{\ell=1}^L$. The procedure is illustrated in Algorithm 2. We then use a hierarchical approach to obtain contrastive explanations for detected change points. First, we identify the feature dimension with the highest feature contribution according to Algorithm 2. We then eliminate the dissimilarity associated with this feature by replacing its values in the current sample with the empirical mean of the same feature in the reference sample. The feature-removal step is validated by recomputing the random projections $S_L$ between the updated sample sets. This validation step indicates whether the reduced samples still contain drifted feature dimensions: under $H_0$, both samples arise from the same underlying process, and the sliced Wasserstein discrepancy between their empirical distributions should approach zero. We propose a stopping criterion based on the norm of the mean difference, which is upper bounded by a constant depending on $d$, $N$, and the covariance matrix. The stopping criterion is derived in Section D.4. Our proposed model-specific explanation procedure is illustrated in Algorithm 3.

The key intuition is that large projected Wasserstein distances identify directions along which the reference and current samples differ most strongly. Since each projection direction is a weighted combination of the original feature dimensions, the weights of the most discrepant projections provide information about which features contribute most to the observed distributional discrepancy. Aggregating these weights over the most informative slices yields a feature-level attribution score. Thus, the method complements change-point detection with an interpretable summary of the feature dimensions most strongly associated with

the detected drift. This attribution should be understood as a contrastive description of the observed distributional change, rather than as a causal explanation. This is particularly useful in high-dimensional settings, where directly inspecting the full multivariate shift is difficult.

We conduct a sensitivity analysis of Algorithm 2 and Algorithm 3 with respect to changes in the dimension and number of samples of the underlying distribution, as well as varying hyperparameters $L$ and $q$. The results are summarized in Section D.5 and support the robustness and adaptivity of the proposed stopping criterion.

---

**Algorithm 1** SWCPD

**Input:** Time series $\boldsymbol{D}$, Window length $\boldsymbol{w}$, Number of projections $\boldsymbol{L}$, Wasserstein order $\boldsymbol{p}$, max AR-lag $\mathbf{K}_{\max}$, quantile threshold $\boldsymbol{q}$

---
1: $\mathcal{D} \leftarrow \text{TIMESERIESDATASET}(D, w)$
2: $C_0 \leftarrow 0$
3: detect $\leftarrow$ True
4: $\mathcal{A}, \mathcal{B}, \mathcal{CP}_{loc} \leftarrow [\,]$
5: **for** $t = 0, 1, \ldots, |\mathcal{D}| - 1$ **do**
6:     $\Theta \leftarrow \text{SAMPLETHETA}(d, L)$
7:     $(X_{\text{ref}}, X_{\text{cur}}) \leftarrow \mathcal{D}[t]$
8:     $S_t \leftarrow \text{PROJECT}(\hat{\mathbb{P}}_{X_{\text{ref}}}, \hat{\mathbb{P}}_{X_{\text{cur}}}, \Theta, p)$
9:     $\ell_t \leftarrow \text{mean}(S_t)$
10:     $C_{t+1} \leftarrow C_t + \ell_t$
11:     **if** $t > 0$ **then**
12:         **if** $C_{t+1} \geq U_{t-1}$ **then**
13:             **if** detect $=$ True **then**
14:                 $\hat{\tau} \leftarrow t + \lfloor w/2 \rfloor$
15:                 Append $\hat{\tau}$ to $\mathcal{CP}_{loc}$
16:                 detect $\leftarrow$ False
17:             **end if**
18:         **else**
19:             detect $\leftarrow$ True
20:         **end if**
21:     **end if**
22:     $(\hat{a}_t, \hat{b}_t) \leftarrow \text{MOMESTIMATES}(S_t)$
23:     Append $\hat{a}_t, \hat{b}_t$ to $\mathcal{A}, \mathcal{B}$
24:     $h \leftarrow \min(K_{\max}, t + 1)$
25:     $\hat{a}_{t+1} \leftarrow \frac{1}{h} \sum_{j=t-h+1}^{t} \mathcal{A}[j]$
26:     $\hat{b}_{t+1} \leftarrow \frac{1}{h} \sum_{j=t-h+1}^{t} \mathcal{B}[j]$
27:     $U_t \leftarrow C_t + \kappa_{t+1}(1 - q)$
28: **end for**
29: **Return** $\mathcal{CP}_{loc}$

---

**Algorithm 2** Calculate Feature Contribution

**Input:** Slices $\mathbf{S_L}$, Projection parameters $\boldsymbol{\theta}$, Wasserstein order: $\mathbf{p}$, upper-tail probability: $\mathbf{z}$

---
1: $S_L^{\rightarrow} = [S_d(\theta_{\pi(1)}), \ldots, S_d(\theta_{\pi(L)})]$       $\triangleright$ Sort
2: $\theta_{1:L}^{\rightarrow} = [\theta_{\pi(1)}, \ldots, \theta_{\pi(L)}]$
3: $i_q \leftarrow \lceil zL \rceil$
4: $I_s = \frac{1}{L - i_q} \sum_{i=i_q}^{L} |\theta_{\pi(i)}|$
5: **Return** $I_s$

---

**Algorithm 3** Hierarchical validated explanations

**Input:** Data: $\mathbf{X}, \mathbf{Y}$, Wasserstein order: $\mathbf{p}$, upper-tail probability: $\mathbf{z}$, Number of projections: $\mathbf{L}$

---
1: cl $\leftarrow [1, \ldots, d]$
2: cr $\leftarrow \emptyset$
3: $C \leftarrow \sqrt{\frac{2}{N} \text{tr}(\Sigma_X)}$
4: **while** $\|D\| \geq C$ **and** $|\text{cl}| > 0$ **do**
5:     Calculate random projections $\mathbf{S}_L$
6:     Calculate $I_s$ (Algorithm 2)
7:     $i_* \leftarrow \arg\max I_s$
8:     cr $\leftarrow \text{add}(i_*, \text{cr})$
9:     $\mathbf{Y}[:, i_*] \leftarrow \mathbb{E}[\mathbf{X}[:, i_*]]$
10:     $D \leftarrow \frac{1}{N} \sum_{i=1}^{N} X_i - \frac{1}{N} \sum_{i=1}^{N} Y_i$
11: **end while**
12: **Return** cr

---

## 5 Experiments

We first evaluate the alignment of feature explanations obtained with Algorithm 3 to popular feature explanation methods. We demonstrate that Algorithm 3 leads to informative insights that enable contrastive explanations for change detection. In the second part of this section, we demonstrate the feasibility of our method against several popular offline and online change-point detection methods, achieving results that are comparable or better in terms of predictive performance and reliability.

Table 1: Mean alignment (Equation (4)) of ground truth features with SWD, IG, GS, and DL explanations for dimensions $d = 10, 20$ and various number of drifted components $k = 1, 3, 7, 9$ over 5 different runs.

| | $d = 10$ | | | | $d = 20$ | | | |
|---|---|---|---|---|---|---|---|---|
| | IG | GS | DL | SWD | IG | GS | DL | SWD |
| $k = 1$ | $\mathbf{0.991 \pm 0.010}$ | $0.987 \pm 0.015$ | $0.982 \pm 0.021$ | $0.961 \pm 0.057$ | $\mathbf{0.999 \pm 0.001}$ | $0.999 \pm 0.000$ | $0.998 \pm 0.002$ | $0.991 \pm 0.003$ |
| $k = 3$ | $0.928 \pm 0.052$ | $0.929 \pm 0.051$ | $0.930 \pm 0.052$ | $\mathbf{0.997 \pm 0.001}$ | $0.958 \pm 0.020$ | $0.957 \pm 0.021$ | $0.959 \pm 0.016$ | $\mathbf{0.991 \pm 0.003}$ |
| $k = 7$ | $0.915 \pm 0.041$ | $0.916 \pm 0.040$ | $0.911 \pm 0.040$ | $\mathbf{0.997 \pm 0.001}$ | $0.933 \pm 0.007$ | $0.933 \pm 0.007$ | $0.934 \pm 0.016$ | $\mathbf{0.994 \pm 0.003}$ |
| $k = 9$ | $0.893 \pm 0.040$ | $0.894 \pm 0.040$ | $0.898 \pm 0.034$ | $\mathbf{0.998 \pm 0.001}$ | $0.913 \pm 0.020$ | $0.913 \pm 0.020$ | $0.908 \pm 0.031$ | $\mathbf{0.994 \pm 0.001}$ |

## 5.1 Interpretability

In our study, we use Integrated Gradients (IG) (Sundararajan et al., 2017), Gradient Shap (GS) (Lundberg & Lee, 2017), and DeepLIFT (DL) (Shrikumar et al., 2017) to obtain baseline feature importance for synthetic data and real-world data.

**Synthetic Data.** We generate data $X_{1:N} \sim \mathcal{N}(\mu_d, \Sigma_d)$ for $N = 5000$ and $d = 10, 20$, with mean $\mu_d$ and covariance $\Sigma_d$. Each component of $\mu_d^i$ follows a normal distribution and is sampled independently. We randomly select $k \leq d$ indices in $\mu_d$ and sample an individual severity $\epsilon_i \sim \mathcal{N}(2, 1)$ for each selected index, which is added to the mean prior to the drift $\tilde{\mu} = \mu + \epsilon$. This ensures that some feature dimensions are more important for the total drift and should show a higher contribution to the explanation scores. We generate data after the drift $\tilde{X}_{1:N} \sim \mathcal{N}(\tilde{\mu}_d, \Sigma_d)$, throughout the experiments, we vary the number of drifted components $k = 1, 3, 7, 9$ and set $\Sigma_d = \mathbb{I}_d$. For a binary classification of samples before and after the drift, we train a simple fully connected neural network with three hidden layers with $128, 64$, and $32$ units, respectively. We use IG, GS, and DL to calculate feature attributions $\phi(X), \phi(\tilde{X})$ for data before and after the drift occurred. For SWD, we follow Algorithm 3 to assign explanation vector $e_{\text{SWD}}$. To quantify how severe the differences in the attribution scores for IG, GS, and DL are, we assign some explanation scores by calculating the absolute differences between both attributions $e := |\phi(X) - \phi(\tilde{X})|$. We use the cosine similarity to quantify the alignment between IG, GS, DL, and SWD explanations and the reference ground truth explanation vectors,

$$s(e, e_{\text{gt}}) = \frac{\langle e, e_{\text{gt}} \rangle}{||e||_2 ||e_{\text{gt}}||_2}. \tag{4}$$

We investigate the alignment for different scenarios by varying $d = 10, 20$ and $k = 1, 3, 7, 9$. For each parameter pair, we simulate data and calculate alignment between ground truth explanations and SWD explanation, IG, GS, and DL for five different runs. In Table 1, we report the average alignment between ground truth explanations and reference explanations obtained by IG, GS, DL, and SWD after the first iteration of Algorithm 3. **Real World Data.** We employ a Vision Transformer (ViT) model (Dosovitskiy et al., 2021) for image classification on the MNIST (LeCun et al., 2010) dataset. Details on the model architecture can be found in Section D.1.1. We simulate a streaming behavior of samples from a particular class, which then abruptly changes to another class. The average feature attribution per class shows the most important features for a given concept, e.g., number 7 has distinct characteristics (edges, curvature) to number 0. However, the general representation of number 1 should be similar to 7 on a feature level, such that the classification model indicates a substantial overlap in the feature attributions. We calculate the absolute differences of the average feature attributions for two classes using IG, GS, and DL, which we use as a qualitative measure to explain the drift. We modify the projection procedure in Algorithm 3 by using the unit vectors to obtain a pixelwise importance and terminate after 250 iterations. We found that for two distinct digits, there are 245.08 pixels on average, which show an absolute deviation above 0.1. Changes below this are generally indistinguishable, such that this reduced set captures the most important pixels which are a valid representation of the original class, therefore 250 is a conservative qualitative stopping criterion. Figure 1 shows the results for three challenging drifts. IG and GS show similar results, which is plausible since GS computes expected gradients and can be seen as an extension of IG. We simulated adversarial attacks on the ViT model using FGSM (Goodfellow et al., 2014) with $\epsilon = 5 \times 10^{-4}$ and compared the average adversarial example to the average non-adversarial example, Figure 1.

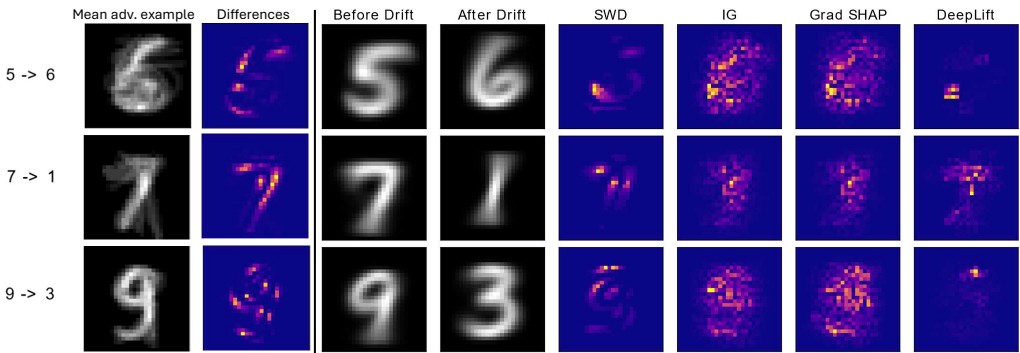

Figure 1: Shows the average adv. example and its corresponding differences for three different drifts (left). On the right-hand side, we see the average example of each class before and after the drift alongside the highlighted feature attributions with SWD, IG, GS, and DL.

## 5.2 Change point detection

In this section, we evaluate our proposed method on one synthetic dataset and four real-world datasets: MNIST, Human Activity Recognition (HAR) (Anguita et al., 2013), Human Activity Segmentation Challenge (HASC) (Ermshaus et al., 2023a), and Occupancy (Candanedo & Feldheim, 2016). MNIST is primarily challenging due to its high dimensionality, whereas the sensor datasets HAR and HASC exhibit changes in both mean and variance. We report Area Under the Curve (AUC), segmentation covering scores (COV), average detection delay (DD), and the average number of false positives (FP). These metrics capture complementary aspects of change point detection performance. Following the evaluation protocol of Van den Burg & Williams (2020) and Ermshaus et al. (2023b), AUC and covering are used as comparative benchmark metrics over the full sequence and are therefore mainly informative from an offline evaluation perspective. In contrast, average detection delay directly reflects the sequential alarm perspective and is the primary metric for assessing online detection performance. The number of false positives is relevant in both settings, as it measures the reliability of the detector and its tendency to raise spurious alarms. For a detailed description and motivation of the evaluation metrics, we refer the reader to Van den Burg & Williams (2020) and Ermshaus et al. (2023b). In our experimental setup, the reported standard deviations are computed across sequences within each benchmark dataset. Occupancy consists of a single sequence in our evaluation, therefore, we report mean and standard deviation across ten different random seeds. Zero standard deviation are due to deterministic behavior.

We compare our method against five popular change point detection methods: BOCPD (Adams & MacKay, 2007), e-divisive (Matteson & James, 2014), KCP (Arlot et al., 2019), OT-CPD (Cheng et al., 2020a), and RuLSIF (Liu et al., 2013a); one time-series segmentation method, ClaSP Ermshaus et al. (2023b); and two deep-learning-based methods, ONNR and ONNC from Hushchyn et al. (2020); Hushchyn & Ustyuzhanin (2021), which we refer to as DeepRuLSIF and DeepCLF, respectively. In general, an appropriate hyperparameter choice consists of a window length $w$ smaller than the average segment length, $K_{max}$ chosen equal to $w$ or as a smaller fraction of $w$ to obtain a more adaptive threshold with a shorter autoregressive lag, Wasserstein order $p \in \{2, 4\}$, a sufficiently large number of projections $L > 500$, and a quantile level $q < 0.15$ for a robust detection threshold.

Table 2: Shows average AUC scores with standard deviation, and average number of false positives and detection delay with min-max values for synthetic data

| | Exponential | | | | | Mixture | | | | |
|---|---|---|---|---|---|---|---|---|---|---|
| | AUC (↑) | | FP (↓) | | DD (↓) | | AUC (↑) | | FP (↓) | | DD (↓) |
| $\lambda$ | $\tau = 10$ | $\tau = 20$ | $\tau = 10$ | $\tau = 20$ | | $\sigma / \lambda$ | $\tau = 10$ | $\tau = 20$ | $\tau = 10$ | $\tau = 20$ | |
| 0.5 | $0.6 \pm 0.13$ | $0.93 \pm 0.13$ | 1.2 (1; 2) | 0.2 (0; 1) | 14.8 (11; 18.5) | 0.25 | $1.0 \pm 0.0$ | $1.0 \pm 0.0$ | 0 (0; 0) | 0.0 (0; 0) | 5.6 (3.5; 7.5) |
| 0.1 | $0.47 \pm 0.1$ | $0.55 \pm 0.17$ | 0.8 (0; 1) | 0.6 (0; 1) | 16.6 (0; 22) | 0.5 | $0.53 \pm 0.16$ | $0.87 \pm 0.16$ | 1.4 (1; 2) | 0.4 (0; 1) | 14.9 (10.5; 20.5) |

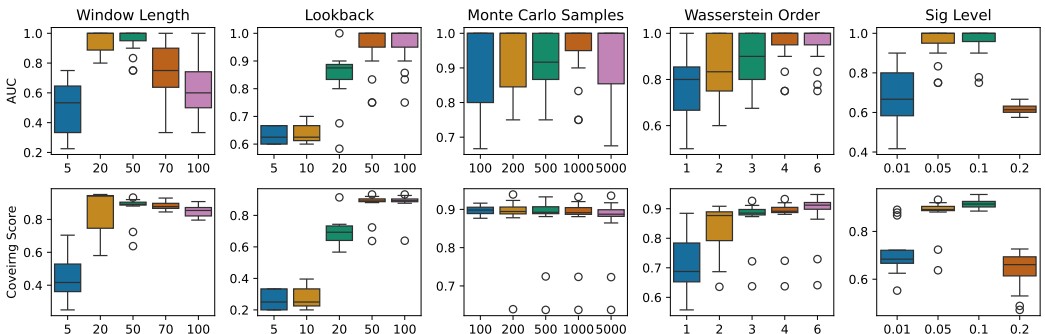

Figure 2: Boxplots of AUC and Covering scores for each parameter variation while keeping the other parameters fixed.

Table 3: Shows the average discriminative accuracy of Algorithm 3 and the influence on the detection ability measured by the change of true positives and covering score.

| $\delta$ | 0.2 | 0.3 | 0.5 | 0.7 | 1.0 | 2.0 |
|---|---|---|---|---|---|---|
| Acc | $0.68 \pm 0.11$ | $0.77 \pm 0.08$ | $0.87 \pm 0.08$ | $0.90 \pm 0.08$ | $0.90 \pm 0.08$ | $0.95 \pm 0.08$ |
| $\Delta_{\mathrm{TP}}$ | $1.0 \pm 0.0$ | $1.0 \pm 0.0$ | $1.0 \pm 0.0$ | $1.0 \pm 0.0$ | $1.0 \pm 0.0$ | $1.0 \pm 0.0$ |
| $\Delta_{\mathrm{Cov}}$ | $0.49 \pm 0.01$ | $0.49 \pm 0.01$ | $0.50 \pm 0.0$ | $0.50 \pm 0.0$ | $0.50 \pm 0.0$ | $0.50 \pm 0.0$ |

**Synthetic Data:** We construct a data stream of $d = 50$ exponential distributions $x_i \sim \mathrm{Exp}(\lambda) + c_i$, where $c_i$ is randomly sampled within $(-3, 3)$ for $i = 1, \ldots, d$. We simulate 3 segments, where each segment consists of 500 samples. We randomly select a total of 3 features for which we inject a drift by offsetting the mean $c_i$ randomly sampled within $(-3, 3)$ for each drifted feature. Additionally, we generated a mixture distribution consisting of 20 Exponential distributions and 30 Gaussian distributions. In Section C.1, we provide a detailed description of the sampling procedure. For all experiments on synthetic data, we set the window length $w = 50$, the lookback window for the estimation of shape- and rate parameters $K_{\max} = 50$, $p = 2$, and $L = 5000$. Table 2 shows the average AUC scores, number of false positives, and detection delay for Exponential- and mixture distributions for different distributional parameters $\lambda, \sigma$, and different margin of errors $\tau$ in the calculation of AUC scores, false positives, see Van den Burg & Williams (2020).

**Faithfulness:** Additionally, we investigate the faithfulness of *discriminative features* derived using Algorithm 3. For this matter, we simulate a 50/50 mixture distribution of Gaussian and Exponential random variables with $d = 50$ with 500 observations. We randomly select 10 features for which we inject a mean shift at $t = 250$ with a magnitude uniformly sampled in $[-\delta, \delta]$. We let our method identify the 10 most discriminative features and mask the time series by removing the identified features. We use an independent oracle (KCP) with an AUC and covering score of 1.0 on the original data, and evaluate it on the masked data. We report the True Positive change $\Delta_{\mathrm{TP}} = \mathrm{TP}_{\mathrm{clean}} - \mathrm{TP}_{\mathrm{masked}}$ and covering change $\Delta_{\mathrm{Cov}} = \mathrm{Cov}_{\mathrm{clean}} - \mathrm{Cov}_{\mathrm{masked}}$. Since, $\mathrm{TP}_{\mathrm{clean}} = 1.0$, the desired $\Delta_{\mathrm{TP}} = 1.0$ which indicates that without the discriminative features, the oracle no longer detects any change point. Thus, the desired covering change is $\Delta_{\mathrm{Cov}} = 0.5$ as no segmentation leads to $\mathrm{Cov} = 0.5$. Additionally, we calculate the discriminative accuracy as the fraction of identified discriminative features relative to the ground-truth discriminative features, results are summarized in Table 3.

**False Positive Control:** We evaluate the false-alarm behavior of SWCPD in a controlled no-change regime using MNIST sequences containing samples from a single digit class only. Since the class label remains fixed throughout each sequence, no semantic change point is present, and all detected change points are counted as false positives. For each digit, we repeat the detection procedure for different adaptive threshold levels $q \in \{0.01, 0.05, 0.10, 0.15, 0.20\}$ and report the mean number of false alarms together with the standard deviation across runs. Figure 3 shows that the false positives decreases as the threshold becomes more conservative, i.e., as $1 - q$ increases.

For lower threshold levels, SWCPD produces a larger number of false alarms and exhibits higher variability across runs. In contrast, for $1 - q \geq 0.95$, the detector produces essentially no false alarms in this no-change regime. These results indicate that the quantile parameter $q$ provides effective control over the sensitivity of the detector, with smaller values of $q$ yielding more conservative and better-calibrated behavior under stable streams.

**Real-world data:** We use three popular datasets commonly used for the evaluation of change point detection, containing sensor measurements over time, Occupancy, HASC, and HAR. Additionally, we sampled 15 sequences containing 600-1000 digits. A detailed description of each dataset and sampling proce-

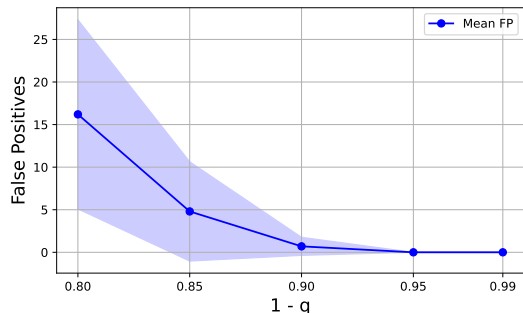

Figure 3: False-alarm behavior of SWCPD on no-change regimes using single-label MNIST streams.

dure applied for MNIST can be found in Section C. We describe the full set of hyperparameters for each method and dataset in Section B.

**SWCPD is the most reliable detection method among popular offline methods.** We report the results in Table 4. Across the considered datasets, SWCPD achieves competitive AUC scores while consistently maintaining a low number of false positives. In particular, SWCPD exhibits strong false-positive control on all datasets and substantially reduces false alarms compared with several traditional baselines, such as KCP on the HASC dataset. The method also achieves favorable detection delays in several settings, most notably on HASC. While other methods obtain better results on some individual metrics, such as COV or DD on specific datasets, SWCPD provides a robust overall trade-off between detection accuracy, detection delay, and false-positive control. However, the lower COV score is mainly due to SWCPD's conservative behavior. Because it is tuned to avoid false positives, it predicts fewer change points than more sensitive baselines. While this reduces spurious detections, it can miss true changes and merge ground-truth segments, lowering COV.

Table 4: Shows the average AUC & Covering scores, average detection delay (DD), and false positives (FP) together with the standard deviation of SWCPD and offline methods over real-world datasets. **Bold** numbers indicate best performance; underlined values are statistically equal to best results [2].

| Dataset | | Method | | | | |
|---|---|---|---|---|---|---|
| | | e-divisive | KCP | CLasP | OT-CPD | SWCPD |
| Occupancy | AUC (↑) | $0.34 \pm 0.0$ | $0.52 \pm 0.0$ | $\underline{0.58} \pm 0.0$ | $0.40 \pm 0.0$ | $\mathbf{0.59} \pm .008$ |
| | COV (↑) | $0.64 \pm 0.0$ | $0.64 \pm 0.0$ | $0.19 \pm 0.0$ | $0.73 \pm 0.0$ | $\mathbf{0.81} \pm .001$ |
| | DD (↓) | $\underline{53} \, (53; 53)$ | $77 \, (77; 77)$ | $- \, (-; -)$ | $129 \, (129; 129)$ | $\mathbf{52} \, (49.5; 52.8)$ |
| | FP (↓) | $12 \, (12; 12)$ | $11 \, (11; 11)$ | $- \, (-; -)$ | $11 \, (11; 11)$ | $\mathbf{4} \, (4; 4)$ |
| MNIST | AUC (↑) | $\underline{0.96} \pm 0.05$ | $0.91 \pm 0.06$ | $0.63 \pm 0.03$ | $0.95 \pm 0.05$ | $\mathbf{0.97} \pm 0.07$ |
| | COV (↑) | $\underline{0.95} \pm 0.05$ | $0.93 \pm 0.05$ | $0.26 \pm 0.06$ | $\mathbf{0.96} \pm 0.10$ | $0.89 \pm 0.07$ |
| | DD (↓) | $9.41 \, (0; 23)$ | $21.7 \, (0; 71)$ | $- \, (-; -)$ | $\mathbf{6.2} \, (0; 26)$ | $11.8 \, (8; 14.5)$ |
| | FP (↓) | $0.4 \, (0; 1)$ | $0.66 \, (0; 2)$ | $- \, (-; -)$ | $0.4 \, (0; 1)$ | $\mathbf{0.13} \, (0; 1)$ |
| HASC | AUC (↑) | $0.73 \pm 0.12$ | $0.66 \pm 0.14$ | $0.84 \pm 0.15$ | $0.79 \pm 0.2$ | $\mathbf{0.87} \pm 0.12$ |
| | COV (↑) | $0.57 \pm 0.19$ | $0.59 \pm 0.32$ | $\mathbf{0.79} \pm 0.18$ | $0.75 \pm 0.25$ | $\underline{0.78} \pm 0.19$ |
| | DD (↓) | $357 \, (0; 1264)$ | $334 \, (0; 1540)$ | $180 \, (0; 1054)$ | $233 \, (0; 1342)$ | $\mathbf{39} \, (0; 688)$ |
| | FP (↓) | $3.8 \, (0; 8)$ | $14 \, (0; 47)$ | $0.78 \, (0; 4)$ | $3.7 \, (0; 18)$ | $\mathbf{0.09} \, (0; 1)$ |
| HAR | AUC (↑) | $0.82 \pm 0.07$ | $\mathbf{0.85} \pm 0.06$ | $0.53 \pm 0.05$ | $0.73 \pm 0.06$ | $\mathbf{0.85} \pm 0.07$ |
| | COV (↑) | $0.76 \pm 0.12$ | $\mathbf{0.82} \pm 0.07$ | $0.11 \pm 0.04$ | $0.52 \pm 0.07$ | $0.56 \pm 0.04$ |
| | DD (↓) | $4.7 \, (1.25; 9.3)$ | $3.7 \, (1.0; 7.7)$ | $10.3 \, (9; 12)$ | $\mathbf{1.8} \, (0.5; 4.2)$ | $4.8 \, (2.8; 6.5)$ |
| | FP (↓) | $4.9 \, (1; 14)$ | $2.5 \, (0; 8)$ | $0.33 \, (0; 1)$ | $0.2 \, (0; 1)$ | $\mathbf{0.1} \, (0; 1)$ |

**SWCPD provides a favorable trade-off compared with popular online and deep learning-based detection methods, achieving consistently low false-positives on average while maintaining**

---

[2]Best performance is determined after applying a paired t-test, bold numbers indicate best absolute performance, underlined numbers indicate equal performance with a smaller reported metric.

Table 5: Shows the average AUC & Covering scores, average detection delay (DD), and false positives (FP) together with the standard deviation of SWCPD and online methods over real-world datasets. **Bold** numbers indicate best performance; underlined values are statistically equal to best results.

| Dataset | | Method | | | | |
|---|---|---|---|---|---|---|
| | | BOCPD | RuLSIF | DeepRuLSIF | DeepCLF | SWCPD |
| Occupancy | AUC ($\uparrow$) | $0.57 \pm 0.0$ | $0.38 \pm 0.0$ | $0.34 \pm .062$ | $0.33 \pm .034$ | $\mathbf{0.59} \pm .008$ |
| | COV ($\uparrow$) | $0.73 \pm 0.0$ | $0.79 \pm 0.0$ | $0.77 \pm .024$ | $0.77 \pm .029$ | $\mathbf{0.81} \pm .001$ |
| | DD ($\downarrow$) | $105 \ (105; 105)$ | $85 \ (85; 85)$ | $100 \ (75; 117)$ | $96 \ (76; 124)$ | $\mathbf{52} \ (49.5; 52.8)$ |
| | FP ($\downarrow$) | $11 \ (11; 11)$ | $8 \ (8; 8)$ | $8 \ (8; 9)$ | $8 \ (7; 10)$ | $\mathbf{4} \ (4; 4)$ |
| MNIST | AUC ($\uparrow$) | $0.69 \pm 0.15$ | $0.63 \pm 0.03$ | $0.91 \pm 0.17$ | $0.93 \pm 0.1$ | $\mathbf{0.97} \pm 0.07$ |
| | COV ($\uparrow$) | $0.78 \pm 0.11$ | $0.26 \pm 0.05$ | $0.92 \pm 0.04$ | $\mathbf{0.94} \pm 0.02$ | $0.89 \pm 0.07$ |
| | DD ($\downarrow$) | $17.8 \ (11; 27)$ | $- \ (-; -)$ | $7.5 \ (3; 23)$ | $\mathbf{6.5} \ (2; 16)$ | $11.8 \ (8; 14.5)$ |
| | FP ($\downarrow$) | $0.93 \ (0; 2)$ | $- \ (-; -)$ | $0.4 \ (0; 2)$ | $0.33 \ (0; 1)$ | $\mathbf{0.13} \ (0; 1)$ |
| HASC | AUC ($\uparrow$) | $0.65 \pm 0.10$ | $0.75 \pm 0.16$ | $0.81 \pm 0.13$ | $\underline{0.85} \pm 0.12$ | $\mathbf{0.87} \pm 0.12$ |
| | COV ($\uparrow$) | $0.66 \pm 0.24$ | $0.66 \pm 0.26$ | $0.75 \pm 0.10$ | $\underline{0.78} \pm 0.13$ | $\underline{0.78} \pm 0.19$ |
| | DD ($\downarrow$) | $445 \ (0; 1866)$ | $559 \ (3.5; 4040)$ | $496 \ (0; 3678)$ | $454 \ (0; 4006)$ | $\mathbf{39} \ (0; 688)$ |
| | FP ($\downarrow$) | $9.0 \ (0; 46)$ | $4.7 \ (0; 24)$ | $1.5 \ (0; 5)$ | $1.3 \ (0; 4)$ | $\mathbf{0.09} \ (0; 1)$ |
| HAR | AUC ($\uparrow$) | $0.76 \pm 0.06$ | $0.72 \pm 0.09$ | $0.81 \pm 0.1$ | $0.80 \pm 0.06$ | $\mathbf{0.85} \pm 0.07$ |
| | COV ($\uparrow$) | $0.53 \pm 0.09$ | $0.54 \pm 0.06$ | $\mathbf{0.67} \pm 0.08$ | $\underline{0.66} \pm 0.07$ | $0.56 \pm 0.04$ |
| | DD ($\downarrow$) | $\mathbf{2.8} \ (1.8; 4.2)$ | $7.1 \ (4.9; 9.1)$ | $3.2 \ (1.1; 6.5)$ | $3.4 \ (1; 5.9)$ | $4.8 \ (2.8; 6.5)$ |
| | FP ($\downarrow$) | $\mathbf{0.1} \ (0; 1)$ | $2.2 \ (0; 4)$ | $0.7 \ (0; 3)$ | $0.8 \ (0; 2)$ | $\mathbf{0.1} \ (0; 1)$ |

**competitive or superior AUC across datasets.** The results from Table 5 demonstrate that SWCPD is competitive with prominent online detection methods, including BOCPD, RuLSIF, DeepRuLSIF, and Deep-CLF. Across the evaluated datasets, SWCPD generally achieves strong AUC performance while maintaining particularly low false-positive. Although deep learning-based methods can obtain comparable or better results on some individual metrics (COV,DD), SWCPD provides a favorable balance between predictive performance and reliability without requiring a learned detection model. Overall, these results suggest that SWCPD is a robust and practical alternative for high-dimensional online change-point detection, especially in settings where false alarms are costly.

## 6  Limitations & Conclusion

Despite the demonstrated effectiveness of SWCPD, some limitations merit attention. First, the reliance on random one-dimensional projections can reduce sensitivity to subtle, local changes in high-dimensional spaces, as these may not always be captured by a limited sampling of directions. Future refinements might involve adaptive or learned projection strategies that more selectively probe feature dimensions most likely to exhibit drift. Second, our adaptive thresholding scheme is motivated by the approximate Gamma behavior of the proposed SW-based slices under suitable assumptions. In practice, however, small sample sizes, heavy-tailed data, or deviations from these assumptions may weaken this approximation and affect threshold calibration. Future work should therefore investigate finite-sample behavior, robustness to heavy-tailed distributions, and alternative data-driven calibration schemes. We introduced SWCPD, a framework for interpretable online change point detection in high-dimensional data streams, leveraging Sliced Wasserstein (SW) distance. By transforming multivariate windows into a one-dimensional score, our method circumvents the computational bottlenecks of traditional CPD techniques. SWCPD combines this score with a Gamma-motivated adaptive quantile threshold and a contrastive explanation module that highlights feature dimensions associated with detected shifts. Across several benchmarks, SWCPD achieves competitive or superior detection performance, with particularly strong false-positive control. The proposed attribution mechanism complements detection by providing interpretable summaries of the observed distributional changes. These results suggest that SWCPD is a promising approach for high-dimensional settings where both reliability and interpretability are important.

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

## Appendix

## A    Ablation study

In the following we are going to investigate the sensitivity and influence of SWCPD for variations in its key hyperparameters. Our proposed method relies on the following hyperparameter:

- $\texttt{L} = 500$: Number of random projections (Monte Carlo samples)

- $\texttt{w} = 50$: Window length

- $\texttt{p} = 2$: Order of Wasserstein distance

- $\texttt{q} = 0.05$ : Significance level

- $\texttt{K}_{\texttt{max}} = k$: Maximum length of lookback window (for moving average calculation)

We conducted experiments using the same MNIST datasets as in the experimental section of the paper, hence the number of change points varies from 2 to 4 with 200 samples for each sub-sequence forming one segment. We defined the following parameter sets, $\texttt{w} \in [5, 20, 50, 70, 100]$, $\texttt{K}_{\texttt{max}} \in [5, 10, 20, 50, 100]$, $\texttt{L} \in [100, 200, 500, 1000, 5000]$, $\texttt{p} \in [1, 2, 3, 4, 6]$, and $q \in [0.01, 0.05, 0.1, 0.2]$. Across all simulation on all 15 datasets, we fixed the random seed for the Monte Carlo samples to obtain reproducible results. We choose the default parameter $\texttt{L} = 5000$, $\texttt{p} = 4$, $\texttt{w} = 50$, $\texttt{K}_{\texttt{max}} = 50$, $q = 0.05$ which we fixed, only varying one parameter within its parameter set respectively. Figure 2 shows the parameter sensitivity of SWCPD for this exemplary dataset. This shows, that the most sensitive parameter are the window length, and lookback window, whereas the number of Monte Carlo samples may be sufficiently large if chosen $\texttt{L} \approx d$. The Wasserstein order should be set above 2, depending on the severity of the drifts, since it amplifies low signals (small distances). The same holds for the significance level as it may be irrelevant if the abrupt changes are significant itself. To further emphasize the influence of the Wasserstein order and significance level, we run additional experiments on synthetic datasets with low drift severities. We used the sampling scheme described in Section C.1, where we set $N = 1500$, $d = 10$ with initial base center $c_0 \in [-4, 4]^{10}$ and 10 different

segments. We selected $\mathcal{V} = \{1, 2, 3\}$ and drift severity was set to $\delta_j \sim \text{Uniform}(-1)$ for each feature index in $\mathcal{V}$. In contrast we sampled the remaining data with i.i.d. Gaussian distribution with mean at each base center respectively and $\sigma = 0.5$ for each component. The result highlights the influence of the significance level for the propagated upper bound as increasing the variable leads to a decrease in the AUC and Covering score since the number of false negatives increases when the upper bound is to close to the cumulative sum. In this example, the Wasserstein order was of secondary importance as changing it lead to similar scores across the datasets, however increasing the Wasserstein order has a smoothing effect on the cumulative sum as small Wasserstein distances nearly vanishes. This can be benefiting for noisy signals. For weak signals, where the abrupt changes are small, we suggest decreasing the Wasserstein order amplifying small changes in the underlying data. Additionally, we performed a Grid Search on MNIST and Occupancy. For both

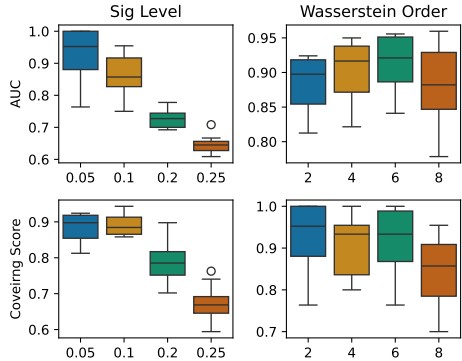

Figure 4: Summary of AUC and Covering scores for varying significance level and Wasserstein order on 10 different synthetic datasets with $d = 10, N = 1500$ and 10 drifts in 3 features simultaneously.

experiments, we fixed $\mathtt{p} = 4, \mathtt{L} = 5000$ while varying the significance level $q$, window size $\mathtt{w}$, and Lookback $\mathtt{K_{max}}$. We limited the possible parameter values for MNIST to $\mathtt{w} \in [20, 30, 40, 50, 100]$, $\mathtt{K_{max}} = [0.5\mathtt{w}, \mathtt{w}]$, and $q = [0.01, 0.05, 0.1]$. We report the average AUC scores for each parameter combination in Figure 5, we see multiple parameter sets achieving high AUC scores. For Occupancy, we limited the possible parameter values to $\mathtt{w} \in [200, 300, 400, 500, 600]$, $\mathtt{K_{max}} = [0.25\mathtt{w}, 0.5\mathtt{w}, 0.75\mathtt{w}, \mathtt{w}]$, and $q = [0.01, 0.05, 0.1]$. We report the AUC scores for each parameter combination in Figure 6, we see multiple parameter sets achieving high AUC scores in comparison to the baseline methods.

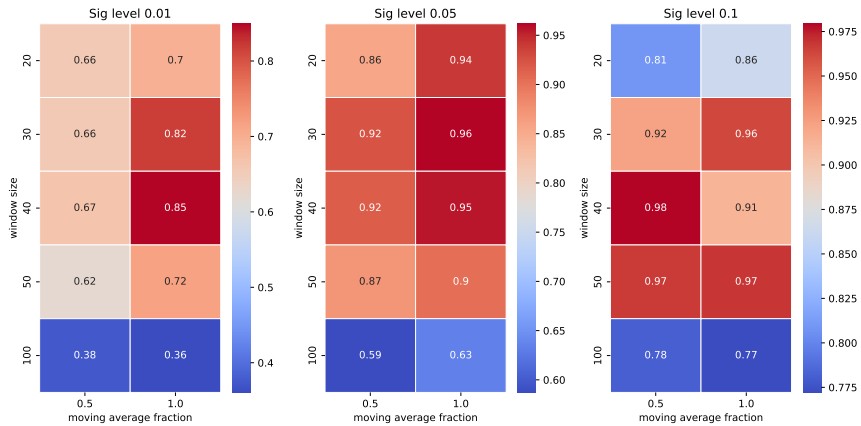

Figure 5: Average AUC scores for various parameter combinations using SWCPD on MNIST sequences.

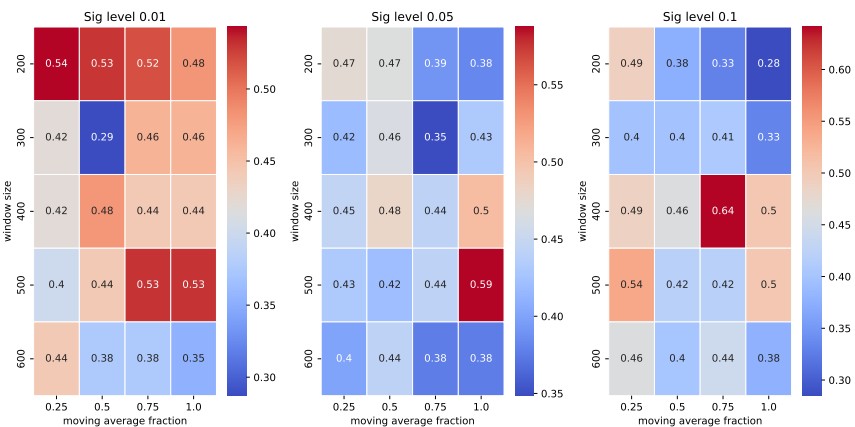

Figure 6: AUC scores for various parameter combinations using SWCPD on Occupancy.

## B    Hyperparameter setting

In the following part, we will describe the reference methods used within the Change Point Detection experiments. Alongside its main parameters and their default values, we also describe the setting for each dataset. We provide an overview of the computational complexity in Table 6. We set the margin or error ($M$ in Van den Burg & Williams (2020)) in the calculation of AUC, Covering, and False Positives to $\tau = 100$ (HASC), $\tau = 10$ (HAR), $\tau = 20$ (MNIST), and $\tau = 10$.(Occupancy).

Table 6: Overview of reference methods and respective time complexity for online and offline change point detection, $K$: number of change points, $d$: dimension, $N$: total samples, $w$: sliding window.

| Method | parametric | non parametric | online | offline | Offline Complexity[3] | Online Complexity[4] |
|---|---|---|---|---|---|---|
| e-divisive | (✓) | | | (✓) | $\mathcal{O}(KN^2)$ | $\mathcal{O}(KN^4)$ |
| KCP | | (✓) | | (✓) | $\mathcal{O}(KdN^2)$ | $\mathcal{O}(KdN^4)$ |
| ClaSP | | (✓) | | (✓) | $\mathcal{O}(KN^2)$ | $\mathcal{O}(KN^4)$ |
| BOCPD | (✓) | | (✓) | | $(-)$ | $\mathcal{O}(Nd)$ |
| OT-CPD | | (✓) | | (✓) | $\mathcal{O}(N(w^3\log(w) + w^2d))$ | $\mathcal{O}(N(w^3\log(w) + w^2d))$ |
| RuLSIF | | (✓) | (✓) | | | $\mathcal{O}(N(w^3 + dw^2))$ |
| SWCPD (ours) | | (✓) | (✓) | | $(-)$ | $\mathcal{O}(N(wdL + Lw\log w))$ |

**SWCPD:**

**Hyperparameter selection.**   For all experiments, hyperparameters were selected according to a fixed set of heuristic rules. In general, we choose the window length $w$ to be smaller than the average segment length, so that the reference and current sub-windows are unlikely to contain multiple change points. The lookback parameter $K_{\max}$ is chosen either equal to $w$ or as a smaller fraction of $w$; smaller values make the adaptive threshold more responsive by using a shorter autoregressive lag. We use Wasserstein orders $p \in \{2, 4\}$, a sufficiently large number of projections $L > 500$, and a quantile level $q < 0.15$, which yields a conservative and robust detection threshold. Dataset-specific values are reported in the following:

- **HASC**
    - $\mathtt{L}, \mathtt{w}, \mathtt{p}, \mathtt{q}, K_{\max} : 500, 500, 2, 0.05, 20$
- **HAR**
    - $\mathtt{L}, \mathtt{w}, \mathtt{p}, \mathtt{q}, K_{\max} : 5000, 20, 2, 0.075, 20$
- **MNIST**

    – $\texttt{L}, \texttt{w}, \texttt{p}, \texttt{q}, \texttt{K}_{\max} : 5000, 50, 4, 0.1, 25$

- **Occupancy**

    – $\texttt{L}, \texttt{w}, \texttt{p}, \texttt{q}, \texttt{K}_{\max} : 1000, 500, 2, 0.05, 500$

**BOCPD (online):** Bayesian Online Change Point Detection (BOCPD) (Adams & MacKay, 2007) is a method used to detect change points in streaming data in real time. It has some desirable properties, such that it can be applied online, is applicable to multivariate data, and quantifies uncertainty (Knoblauch & Damoulas, 2018). The underlying concept of this approach is to monitor the probability of a change point occurring at each time step by maintaining and updating the posterior distribution over potential segmentations of the data. It assumes that data within a segment follows a consistent probabilistic model (e.g., Gaussian), and a change point indicates a shift in the underlying model. There exist many implementation, we use the implementation that comes with the `ocp` package (Pagotto, 2019). The key parameters for this method are:

- `prob_model`: the underlying probability model of the posterior distribution

- `init_params`: the initial parameters for the probability model consiting of $m, k, a, b$

- `hazard_function`: normally set to a constant function with certain hazard rate $\lambda$

We run the experiments with the following parameter sets:

- **HASC**
  - `prob_model` : *"gaussian"*
  - `init_params` : $m = 0, k = 10, a = 0.1, b = 0.01$
  - `hazard_function` : type=constant, $\lambda = 100$

- **HAR**
  - `prob_model` : *"gaussian"*
  - `init_params` : $m = 0, k = 0.01, a = 0.01, b = 1e - 4$
  - `hazard_function` : type=constant, $\lambda = 100$

- **MNIST**
  - `prob_model` : *"gaussian"*
  - `init_params` : $m = 0.3, k = 0.01, a = 0.01, b = 1e - 4$
  - `hazard_function` : type=constant, $\lambda = 100$

- **Occupancy**
  - We additionally applied z-score normalization of the data beforehand to obtain a reasonable distributional setting and obtain change points
  - `prob_model` : *"gaussian"*
  - `init_params` : $m = 0, k = 0.01, a = 0.01, b = 1e - 4$
  - `hazard_function` : type=constant, $\lambda = 100$

**E-divisive (offline):** The e-divisive combines binary bisection together with a permutation test based on an energy divergence measure (Matteson & James, 2014). It is a non-parametric offline change point detection method for multivariate data, making it applicable to a wide range of complex data. We use the implementation from the `ecp` package (Nicholas A. James et al., 2019). The method relies on the following parameters with default specification:

- $\texttt{R} = 199$ : specifies the number of permutations test applied

- $\texttt{sig.lvl} = 0.05$ : the significance level of the permutation test

- $\texttt{min.size} = 30$ : the minimum observations between two subsequent change points

We run the experiments with the following parameter sets:

- **HASC:** $\texttt{R} = 199$, $\texttt{sig.lvl} = 0.05$, $\texttt{min.size} = 500$

- **HAR:** $\texttt{R} = 199$, $\texttt{sig.lvl} = 0.05$, $\texttt{min.size} = 30$

- **MNIST:** $\texttt{R} = 199$, $\texttt{sig.lvl} = 0.05$, $\texttt{min.size} = 30$

- **Occupancy:** $\texttt{R} = 30$, $\texttt{sig.lvl} = 0.05$, $\texttt{min.size} = 400$

**KCP (offline):** Kernel change-point detection (KCP) transforms the data into a RKHS with an associated kernel, which is used to calculate the dissimilarity (cost). The goal is to obtain an optimal segmentation of the input data in the sense of a minimized averaged cost within each segment obtained Arlot et al. (2019). An efficient implementation of this method can be found in Truong et al. (2020), we assume that the number of change points is unknown, hence we rely on `KerneCPD` with `PELT`. The methods relies on the following parameter:

- $\texttt{kernel} = "linear"$: specifies the kernel, cost function

- $\texttt{min\_size} = 1$: minimum segmentation length

- $\texttt{pen}$: penalty or regularization of number of change points identified

The penalty value needs to be specified if the number of change point is unknown. Usually a higher value will lead to fewer change points identified, while a lower value encourages the method to annotate more change point with a more fine grained segmentation. We used the following parameter settings:

- **HASC:** $\texttt{kernel} = "rbf"$, $\texttt{min\_size} = 2$, $\texttt{pen} = 10$

- **HAR:** $\texttt{kernel} = "rbf"$, $\texttt{min\_size} = 2$, $\texttt{pen} = 1$

- **MNIST:** $\texttt{kernel} = "rbf"$, $\texttt{min\_size} = 2$, $\texttt{pen} = 1$

- **Occupancy:** $\texttt{kernel} = "rbf"$, $\texttt{min\_size} = 2$, $\texttt{pen} = 50$

**ClaSP (offline):** ClaSP (Classification Score Profile) is a self-supervised time series segmentation method (Ermshaus et al., 2023b). The implementation is available at `https://github.com/ermshaua/claspy`. It is a dynamic windowing approach which creates a binary classification problem across different split points of the time series using $k$-Nearest Neighbors (k-NN) which is evaluated using corss validation. The score obtained from k-NN is used to evaluate the similarity of both segments, where higher scores indicate a stronger dissimilarity. The main parameters to choose are:

- $\texttt{windwo\_size} = "suss"$: size of the sliding window, default Summary Statistics Subsequence (suss)

- $\texttt{k\_neighours} = 3$: number of nearest neighbours for k-NN

- $\texttt{distance} = "znormed\_euclidean\_distance"$: distance used for k-NN

We used the following parameters:

- **HASC:** $\texttt{windwo\_size} = 50$

- **HAR:** $\texttt{windwo\_size} = 30$

- **MNIST:** $\texttt{windwo\_size} = 100$

- **Occupancy:** $\texttt{windwo\_size} = 30$

**OT-CPD (offline):** OT-CPD (Cheng et al., 2020a) is a optimal transport based change point detection method which calculates the Wasserstein distance between two sliding windows. After obtaining all available data, it applies a matched filter on the Wasserstein test statistic to obtain a more persistent test statistic reducing false positives. OT-CPD annotates a change if the filtered test statistic exceeds a pre-defined threshold. In our experiments, we relied on the implementation available at `https://github.com/kevin-c-cheng/OtChangePointDetection/tree/master`. The main parameters for the change point detection method to choose are:

- `window`: size of the sliding window

We used the following parameters:

- **HASC:** $window = 1000$
- **HAR:** $window = 25$
- **MNIST:** $window = 150$
- **Occupancy:** $window = 750$

**RuLSIF:** Relative unconstrained least-squares importance fitting (RuLSIF) estimates a relative density ratio that mixes the two distributions using a parameter $\alpha$. The relative ratio is approximated using a kernel model, and its parameters are obtained by solving a simple least-squares problem with a closed form solution. From this estimated ratio, the method computes a divergence score that becomes large when the two windows differ. In a sliding window approach this scores is computed for which peaks indicate change points. The main parameters for the change point detection method to choose are:

- $\alpha$: mixture coefficient in $\alpha$-relative density ratio
- `window`: size of the sliding window
- `kernel_num`: number of kernels used
- `steps`: stride of sliding window

We used the following parameters:

- **HASC:** $window = 200$, $\alpha = 0.1$, `kernel_num` $= 10$
- **HAR:** $window = 20$, $\alpha = 0.1$, `kernel_num` $= 10$
- **MNIST:** $window = 100$, $\alpha = 0.1$, `kernel_num` $= 10$
- **Occupancy:** $window = 250$, $\alpha = 0.1$, `kernel_num` $= 10$

**DeepRuLSIF:** DeepRuLSIF (Hushchyn et al., 2020; Hushchyn & Ustyuzhanin, 2021) follows the framework of RuLSIF where the $\alpha$ relative density ratio is estimated using a deep neural network. We rely on the implementation given by [5]. The main parameter for the change point detection method which we varied where:

- `lag_size` : the gap between batches

All other hyperparameter were kept as default. We used the following parameters:

- **HASC:** $lag = 250$

---

[5]`https://gitlab.com/lambda-hse/change-point/online-nn-cpd`

- **HAR:** $\text{lag} = 20$

- **MNIST:** $\text{lag} = 100$

- **Occupancy:** $\text{lag} = 250$

**DeepCLF:** This method trains a neuronal network to distinguish a reference window from a more test window based on a divergence metric. By sliding the windows forward in time and measuring their divergence, peaks in the score curve reveal where the underlying data distribution has changed Hushchyn et al. (2020). The main parameter for the change point detection method which we varied where:

- `lag_size` : the gap between batches

All other hyperparameter were kept as default. We used the following parameters:

- **HASC:** $\text{lag} = 250$

- **HAR:** $\text{lag} = 20$

- **MNIST:** $\text{lag} = 100$

- **Occupancy:** $\text{lag} = 250$

## C  Data

This section describes the datasets used to evaluate the online/offline Change Point Detection methods. We consider one synthetic dataset and four real-world datasets. We have empirically checked the validity of Assumptions (A1)–(A4) in the experimental settings considered in this study.

(A2) and (A4) are standard assumptions in the analysis of random projections. In particular, (A2) is motivated by the spherical central limit theorem, according to which one-dimensional random projections of high-dimensional data are approximately Gaussian under mild regularity conditions. To empirically assess the adequacy of this approximation in our setting, we report additional results in Table 12 for different combinations of the number of projections $L$ and the ambient dimension $d$. These results indicate that the Gaussian approximation is stable across the considered configurations.

Assumptions (A1) and (A3) are more restrictive, as they impose conditions directly on the empirical distributions and their projected distances. Specifically, (A1) may be undermined in practice by heavy-tailed or strongly correlated high-dimensional data. Nevertheless, they are satisfied for the empirical datasets considered in our experiments. The following table provides an empirical verification of these assumptions across all datasets used in the study. This supports the practical relevance of the theoretical conditions and indicates that, although (A1) and (A3) may not hold universally, they are not violated in the experimental regimes considered here.

| Dataset | $\mathbb{E}[||X||]^3 < \infty$ | $||\Sigma_P||_{\text{op}} \leq C$ |
|---|---|---|
| Occupancy | 3.73 (✓) | 0.004 (✓) |
| MNIST | 0.10 (✓) | 20.2 (✓) |
| HASC | 38.1 (✓) | 247 (✓) |
| HAR | 5.05 (✓) | 64.4 (✓) |

Table 7: Validity check of assumptions (A1) and (A3).

We report the maximum value of each sliding window and time series (most conservative empirical bound).

---

[4]Complexity for offline change point detection for a multivariate time series with $d$ dimensions and $N$ observations

[5]Accrued complexity for change point detection at time step $t = N$ for a multivariate time series with $d$ dimensions and in total $N$ observations

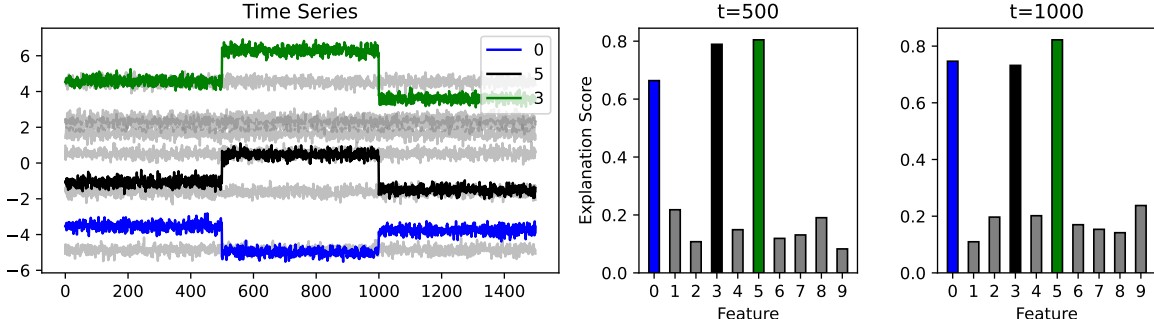

Figure 7: Interpretable change points obtained with SWCPD. Two right plots show feature attributions obtained using Algorithm 3, showing alignment with ground truth root causes of the drifts.

## C.1 Synthetic Data

The proposed sampling scheme generates synthetic data with customizable cluster centers and variable feature dimensions. The process begins by defining an initial base center $\mathbf{c}_0 \in \mathbb{R}^d$, where $d$ is the number of features. This base center serves as the reference point for all subsequent cluster centers.

To generate additional cluster centers, a perturbation process is applied to $\mathbf{c}_0$. Specifically, for each new cluster center $\mathbf{c}_i$, $i = 1, \ldots, k-1$, the following transformation is applied:

$$c_{i,j} = \begin{cases} c_{0,j} + \Delta_j & \text{if } j \in \mathcal{V}, \\ c_{0,j} & \text{otherwise}, \end{cases}$$

where $c_{i,j}$ is the $j$-th feature of the $i$-th cluster center, $\mathcal{V} \subseteq \{1, 2, \ldots, d\}$ is the set of varying feature indices, and $\Delta_j \sim \text{Uniform}(-\delta, \delta)$ is a random offset sampled from a uniform distribution with range $[-\delta, \delta]$.

The sampling process ensures that only the features indexed by $\mathcal{V}$ are modified, while other features remain constant across all cluster centers. After generating the cluster centers, the data points are sampled from a multivariate Gaussian distribution. For each cluster $i$, the samples $\mathbf{x}_i^{(n)}$, $n = 1, \ldots, N_i$, are drawn as:

$$\mathbf{x}_i^{(n)} \sim \mathcal{N}(\mathbf{c}_i, \Sigma),$$

where $\Sigma \in \mathbb{R}^{d \times d}$ is the covariance matrix (diagonal for simplicity) and $N_i$ is the number of samples assigned to cluster $i$. The total number of samples $N$ is distributed evenly across clusters, i.e., $N_i = N/k$.

This scheme allows for precise control over the features that vary between groups $\mathcal{V}$, the degree of variation $\delta$, and the variance of data points within each cluster with $\Sigma$. By adjusting these parameters, synthetic datasets can be tailored for specific experimental purposes, such as evaluating clustering algorithms or analyzing feature-specific effects. In Table 8 we report AUC scores for different variances and drift severities for Gaussian synthetic data with $d = 10$ and 1500 samples with 3 segments. Additionally, Figure 7 illustrates the contrastive explanations for the obtained change points by SWCPD. We set the window length $w = 50$, the lookback window for the estimation of shape- and rate parameters $K_{\max} = 50$, $p = 2$, and $L = 5000$.

Table 8: AUC for different variances $\sigma^2$ and drift severity $|\delta|$

| Source | Value | $\tau = 5$ | $\tau = 10$ | $\tau = 20$ |
|---|---|---|---|---|
| | 0.1 | $1.0 \pm 0.0$ | $1.0 \pm 0.0$ | $1.0 \pm 0.0$ |
| Variance ($\sigma^2$) | 0.5 | $0.8 \pm 0.28$ | $0.93 \pm 0.14$ | $1.0 \pm 0.0$ |
| | 1.0 | $0.65 \pm 0.32$ | $0.75 \pm 0.29$ | $0.91 \pm 0.13$ |
| | 1 | $0.4 \pm 0.15$ | $0.6 \pm 0.26$ | $0.94 \pm 0.08$ |
| Drift Severity ($|\delta|$) | 2 | $0.6 \pm 0.22$ | $0.8 \pm 0.27$ | $0.97 \pm 0.06$ |
| | 3 | $0.71 \pm 0.28$ | $0.87 \pm 0.24$ | $0.98 \pm 0.05$ |

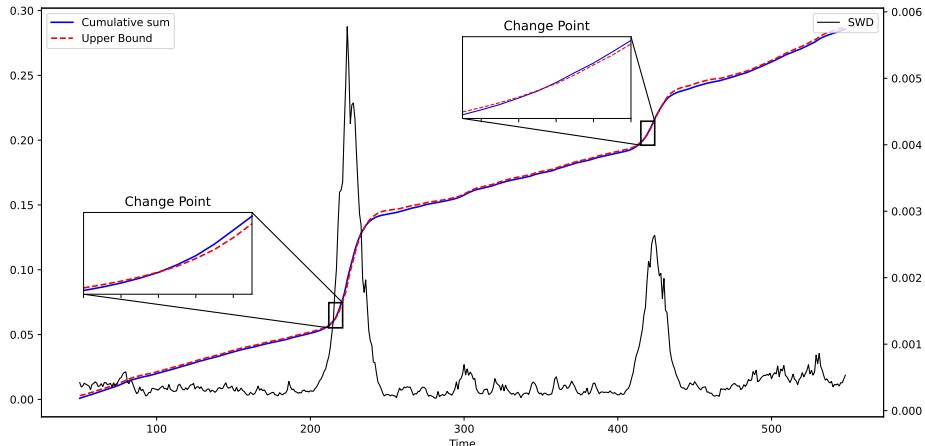

Figure 8: Visualizes our proposed detection method for MNIST data with two change points at $t = 200, 400$. Change points are indicated when the cumulative sum exceed the upper bound which is derived based on past SWDs.

## C.2 MNIST

In order mimic a streaming behaviour, we uniformly sample an initial class (without replacement) and select $K$ instances from the current class. We repeat this procedure and annotate the samples to introduce abrupt changes. Within the scope of the experiments for this paper, we generated 5 distinct data sequences with $2, 3,$ and 4 change points, where each class has 200 samples. We illustrate SWCPDs detection procedure for a sampled MNIST sequence with two change points at $t = 200, 400$ in Figure 8. By calculating tracking the SW distance using a rolling window of $k = 50$ observations, we obtain a one-dimensional signal with two significant spikes at $t_1 = 225$ and $t_2 = 425$ since the within similarity of the rolling window will be the largest when the first half samples belong to class prior to the drift and the second half to the class after the drift. We see, that using a propagated upper bound given the current state instead of purely relying on the distance as a signal, we can anticipate changes more reliable and faster. Moreover, the upper bound is adaptive such that there is no fine tuning or manually shifting the rolling window involved. SWCPD is based on the Sliced Wasserstein distance which is a metric from Optimal Transport (OT). To contextualize the computational performance of our proposed method for other OT-based detection methods such as OT-CPD, and e-divisive, we report the average wall-clock time and standard deviation in Table 9.

Table 9: Runtime comparison of SWCPD and OT-based CPD methods

(a) Average runtimes and AUC scores for OT-baseline methods

| Method | Runtime (s) | AUC |
|--------|-------------|-----|
| OT-CPD | $425 \pm 150$ | $0.95 \pm 0.05$ |
| e-divisive | $5.9 \pm 3.1$ | $0.96 \pm 0.05$ |

(b) Average runtimes and AUC scores of SWCPD for different numbers of projections $L$

| $L$ | Runtime (s) | AUC | vs. OT-CPD | vs. e-divisive |
|-----|-------------|-----|------------|----------------|
| 100 | $1.02 \pm 0.2$ | $0.87 \pm 0.1$ | $+41,979\%$ | $+478\%$ |
| 500 | $2.81 \pm 0.6$ | $0.95 \pm 0.1$ | $+15,024\%$ | $+109\%$ |
| 1000 | $3.33 \pm 0.74$ | $0.95 \pm 0.1$ | $+12,662\%$ | $+77\%$ |
| 5000 | $6.21 \pm 1.3$ | $0.97 \pm 0.07$ | $+6,743\%$ | $-5\%$ |

## C.3 HAR

HAR (Anguita et al., 2013) was collected from 30 volunteers who performed six daily activities (walking, sitting, etc.) while wearing a smartphone on their waist recording various measurements at 50 Hz. Naturally, the change points are given when an activity changes. In total, there are 10.299 observation of $d = 561$ features.

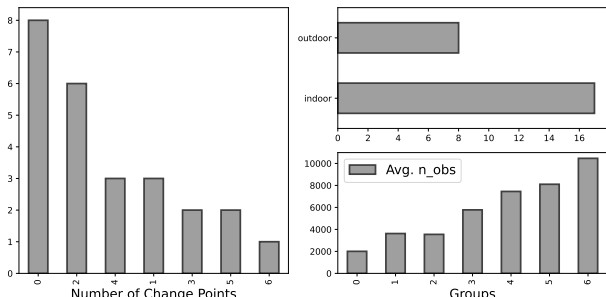

Figure 9: Summary of the data used for the change point detection experiments of HASC dataset.

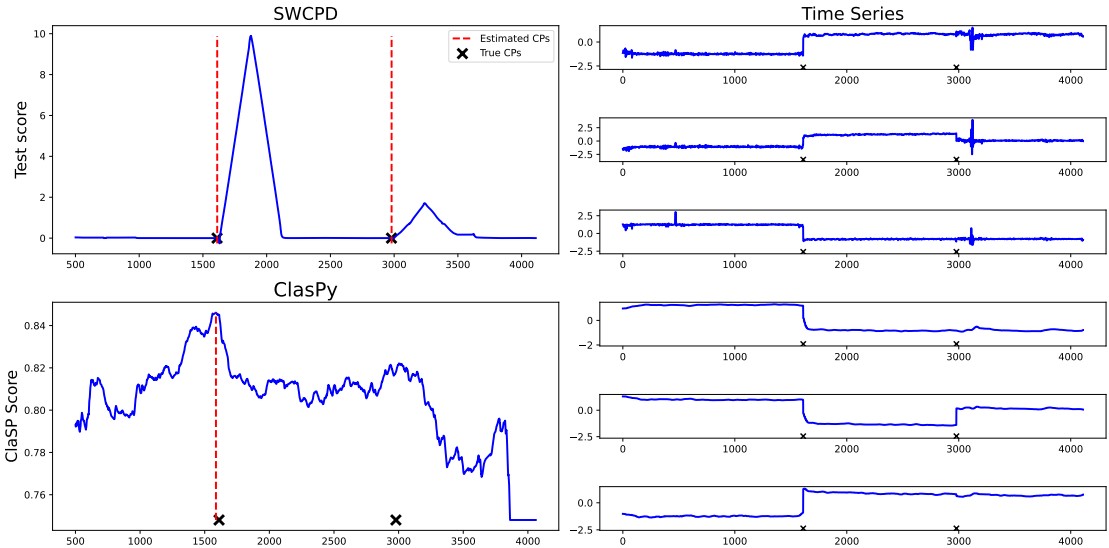

Figure 10: Comparison of Test scores obtained using SWCPD and ClaSP on subject number 243 (left hand side), and corresponding time series (right hand side).

## C.4  HASC

The dataset consists of distinct multimodal multivariate time series monitoring human motion of different daily activities. The data was collected as part of the Human Activity Segmentation Challenge (Ermshaus et al., 2023a) using built-in smartphone sensors. In total, the dataset has 250 time series consisting of 12 different measurements sampled at 50 Hz, where the ground truth change points were independently annotated using video and sensor data. We selected 25 instances covering 17 indoor and 8 outdoor activities for various numbers of segments ranging from 1 to 6. We selected 8 instances with one segment, thus zero change points to asses the sensitivity and robustness of each method when the unknown underlying distribution does not change over time. Furthermore, we see that the average number of observations increases with more segments in the selected data see Figure 9. We specifically considered instances with a single segment to assess each method's robustness to false positives. Figure 10 illustrates the time series of an outdoor activity of a person. In this case, the person is performing three different stretches (standing adductor left, squat stretch for adductors, hamstring stretch right) Figure 11 shows AUC scores of our proposed method and baseline methods for five different annotation margins $\tau \in [25, 50, 100, 150, 200]$, such that if the annotated change point is at least $\tau$ instances away, it is classified as true positive thus contribution to the AUC score. We see that SWCPD shows superior AUC scores for any $\tau$, see Figure 11.

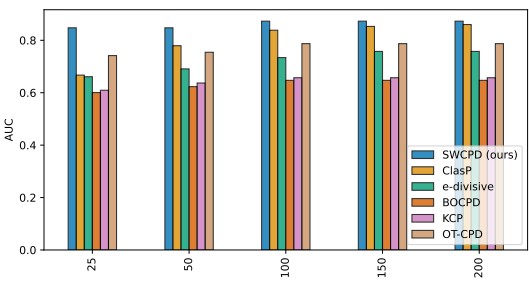

Figure 11: Shows average AUC scores for proposed method and baseline methods on the selected HAR data for different annotation margins $\tau$.

### C.5 Occupancy

This dataset is commonly used for the evaluation of change point detection methods (Van den Burg & Williams, 2020). Originally, it was introduced in (Candanedo & Feldheim, 2016) and captures four different measurements: 1) temperature, 2) humidity level, 3) light, and 4) $CO_2$.

SWCPD is based on the Sliced Wasserstein distance which is a metric from Optimal Transport (OT). To contextualize the computational performance of our proposed method for other OT-based detection methods such as OT-CPD, and e-divisive, we report the average wall-clock time and standard deviation in Table 10.

Table 10: Runtime comparison of SWCPD and OT-based CPD methods

(a) Average runtimes and AUC scores for OT-baseline methods

| Method | Runtime (s) | AUC |
|---|---|---|
| OT-CPD | $96.2 \pm 0.23$ | $0.41 \pm 0.00$ |
| e-divisive | $175.3 \pm 0.19$ | $0.34 \pm 0.00$ |

(b) Average runtimes and AUC scores of SWCPD for different numbers of projections $L$

| $L$ | Runtime (s) | AUC | vs. OT-CPD | vs. e-divisive |
|---|---|---|---|---|
| 100 | $28.2 \pm 0.8$ | $0.48 \pm 0.0$ | $+241\%$ | $+519\%$ |
| 500 | $59.4 \pm 1.25$ | $0.58 \pm 0.0$ | $+62\%$ | $+195\%$ |
| 1000 | $66.6 \pm 1.55$ | $0.59 \pm 0.0$ | $+45\%$ | $+163\%$ |

## D   Additional Experiments

All experiments were conducted on a machine equipped with an AMD Ryzen 7 5700X CPU, 32 GB of RAM, and a RTX 3060 GPU.

### D.1   Explainability

#### D.1.1   MNIST

**Vision Transformer.** We employ a Vision Transformer (ViT) model for image classification on the MNIST dataset. The model processes input images of size $28 \times 28$ pixels, which are divided into non-overlapping patches of size $4 \times 4$, resulting in 49 patches. Each patch is linearly embedded into a 64-dimensional feature space. The transformer consists of 6 layers, each employing multi-head self-attention with 8 heads and a feed-forward network with a hidden dimension of 128. We apply a dropout rate of 0.1 during the embedding and transformer layers to prevent overfitting. Since MNIST images are grayscale, the model is configured to accept single-channel input. The data was split into 90% training set of which 10% into the validation set, while we used the additional 10% for testing. We use Adam with $\lambda = 0.001$ for training over 15 epochs with a batch size of 64.

Table 11: Parameter setting ViT

| Batch size | Epochs | LR | PatchSize | dim | depth | heads | MLP |
|---|---|---|---|---|---|---|---|
| 64 | 15 | $1 \times 10^{-4}$ | 4 | 64 | 6 | 8 | 128 |

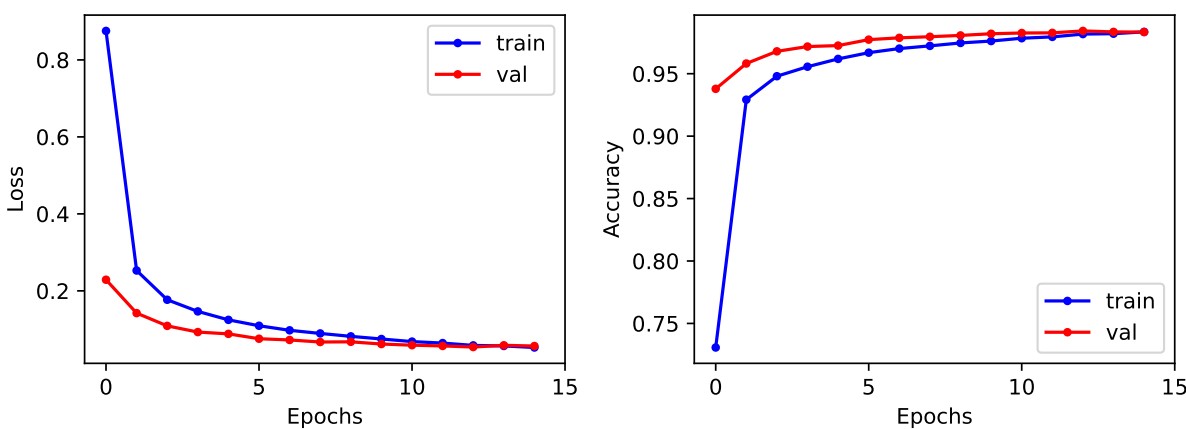

Figure 12: Illustrates Train and validation curves of loss and accuracy over 15 epochs for ViT model.

**CNN.** We use a simple LeNet-5 (LeCun et al., 1998) as a benchmark CNN to investigate model explanations under drifts on MNIST. We use the same train-test split as for the ViT model and Adam optimizer with step size $\lambda = 0.001$. We repeat the same procedure as for the ViT and introduce drifts and investigate the differences in the feature attrituions using SWD, and SoTA explanations methods IG, GS, and DL. From Figure 13, we see that all reference methods align with feature attributions, and hence show the same pattern for differences of before and after drift. Although, all explanation methods align with the most significant feature changes, the pixelwise distance based approach (SWD) narrows them down the most. This can also be seen in Figure 14, which highlights the differences of adversarial examples changing the model output between two given classes, as SWD shows a strong alignment.

### D.2 Uncertainty quantification

We investigate the asymptotic behaviour of the confidence intervals obtained by Proposition E.1 for $X \sim \Gamma(2, 1)$ for various sample sizes and calculate the average confidence intervals for 30 different random samples $X_n$ with sample size $n$. For an increasing sample size, the confidence intervals for both parameters shrinks and is centered around the true parameters as expected since sample mean and variance are consistent, see Figure 15.

### D.3 Distribution of random projections

For the numerical study of the distribution of $w_2^2(\theta) : \theta \mapsto W_2(\mathbb{P}^\theta, \mathbb{Q}^\theta)$, we consider two sample sets $X, Y$ each consisting of 200 MNIST samples with gray-scaled images from the same class respectively. For this example we set the class of each sample from $X$ to 1, and $Y$ to 7. We calculated the SWD between both samples for different numbers of random projections ranging from $L = 100, 500, 1000, 5000$. We then constructed the MoM esitmates of a Gamma distribution based on the set of random projection obtained. Furthermore, we calculated a Kernel density estimation for the random projections itself. This shows that using a Gamma distribution indeed fits the data obtained. Additionally, we complared the sampled quantiles and the theoretical quantiles of the random projections and MoM fitted Gamma distribution to asses the goodness of fit. The result is summarize in Figure 16, as expected, we see that as the number of projection increases, we obtain a better fit. While Figure 16, shows the asymptotic behaviour given by Corollary 3.2

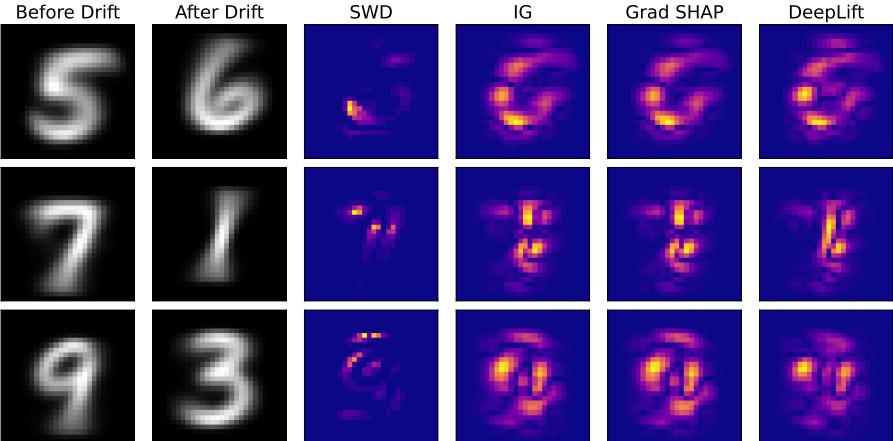

Figure 13: Shows the absolute difference of mean feature attributions for three different drifts and reference methods IG, GS, and DL.

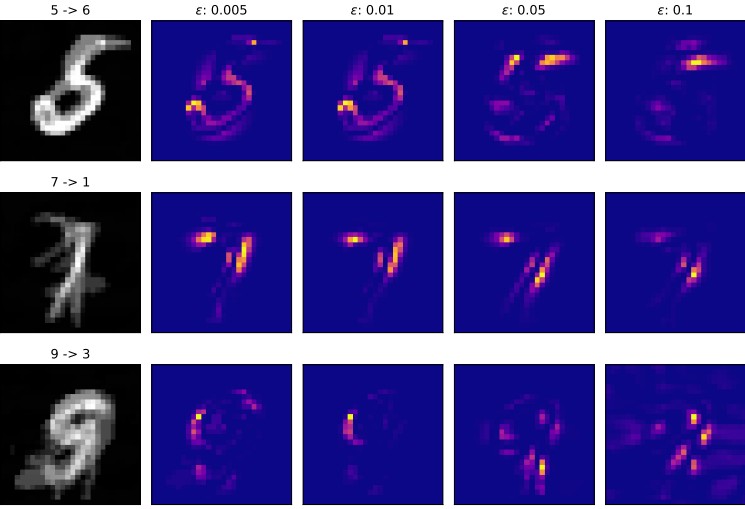

Figure 14: Shows mean adversarial examples (left) which changes the model (CNN) output from $5 \rightarrow 6$, $7 \rightarrow 1$, and $9 \rightarrow 3$ using FGSM for different $\epsilon$, and $L_4$-norm between mean adversarial example and non-adversarial example

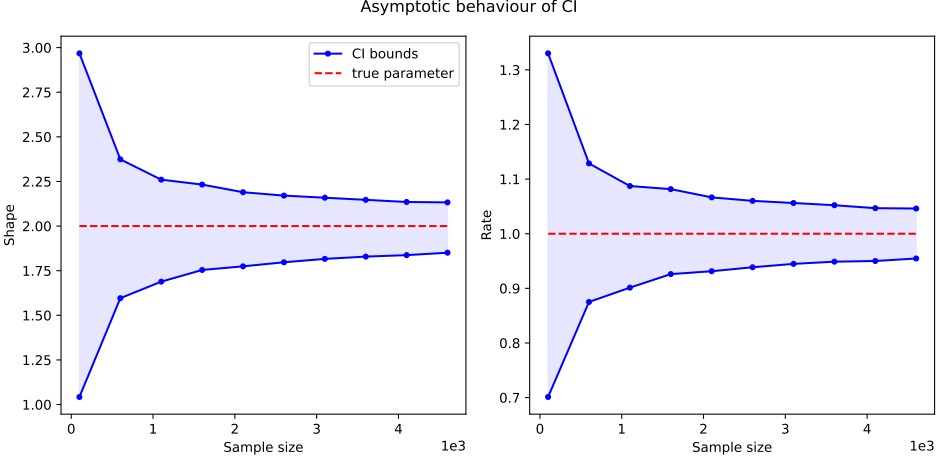

Figure 15: Shows the lower and upper bound of confidence interval (Equation (5)) for MoM estimator $\hat{\alpha}, \hat{\beta}$ averaged over 30 experiments for equidistant sample sizes from $n = 100, \ldots, 5000$.

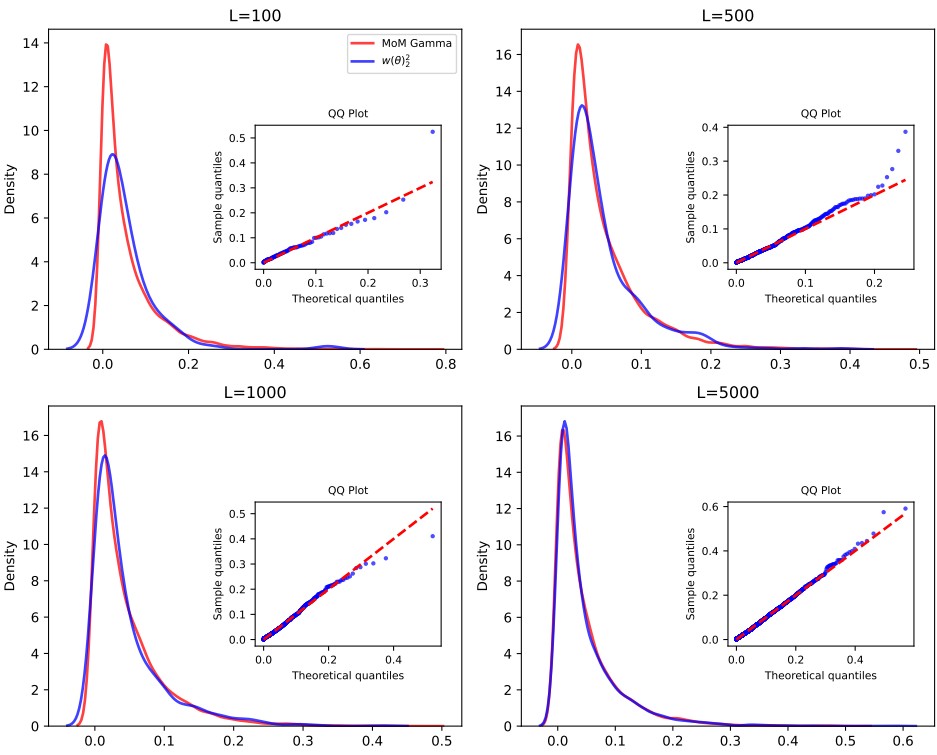

Figure 16: Shows a Kernel density estimation of a gamma density using the MoM estimated parameters (red line) for the random projection for various number of projections $L = 100, 500, 1000, 5000$, and the KDE of random projections (blue line) itself between two samples from MNIST.

Table 12: Average $p$-values obtained using Sharpio-Wilk test

|      |       | $L$ |      |
| :--- | :--- | :--- | :--- |
| $d$  | 100 | 500 | 1000 |
| 10   | 0.44 (✓) | 0.065 (✓) | 0.005 (-) |
| 20   | 0.5 (✓) | 0.3 (✓) | 0.2 (✓) |
| 30   | 0.5 (✓) | 0.4 (✓) | 0.3 (✓) |
| 60   | 0.5 (✓) | 0.5 (✓) | 0.5 (✓) |
| 100  | 0.5 (✓) | 0.5 (✓) | 0.5 (✓) |

of the linear random projections of the Sliced Wasserstein distance, we observed that it also holds for lower-dimensional data, e.g. simulated synthetic data. Consider $x \in \mathbb{R}^d$, we fix a projection direction $\theta_l \sim \mathcal{U}(S^{d-1})$ and consider a sample set $X = (x_1, x_2, \ldots, x_n)$. We set $z_l = \langle X, \theta_l \rangle$, where $z_l$ is normal due to the CLT for $d \to \infty$. We simulated $x$ according to $d$ independent exponential distributions $\lambda = 1$ and applied the Sharpio-Wilk test (Shapiro & Wilk, 1965) to asses wheter the projected samples can be considered normal distributed. In Table 12, we report the average $p$-values projections obtained using $L \in [100, 500, 1000]$ for various dimensions $d$.

**Approximation Error:** We now report the Mean Absolute Error (MAE) between the theoretical quantiles and observed quantiles. The theoretical quantiles are derived from a Gamma distribution based on the MoM estimates from the random projections involved in the calculation of the Sliced Wasserstein distance. The observed quantiles are calculated based on the empirical distribution of the random projections. For each dimension, we simulate two independent datastreams, each consisting of $d$ independent Gaussian distributions with a uniformly sampled mean. We vary $d$ and $L$, use fixed random seeds, and report the results for 10 trials in Table 13.

Table 13: Shows MAE between theoretical and observed quantiles of a Gamma distribution derived from 2-Wasserstein distance between random projections.

| $d$ | $L = 100$ | $L = 1.000$ | $L = 10.000$ |
| :--- | :--- | :--- | :--- |
| 5   | $1.12 \pm 0.17$ | $0.35 \pm 0.03$ | $0.37 \pm 0.01$ |
| 10  | $0.345 \pm 0.06$ | $0.31 \pm 0.06$ | $0.15 \pm 0.01$ |
| 20  | $0.401 \pm 0.07$ | $0.16 \pm 0.03$ | $0.02 \pm 0.01$ |
| 100 | $0.291 \pm 0.05$ | $0.11 \pm 0.01$ | $0.04 \pm 0.01$ |
| 200 | $0.298 \pm 0.05$ | $0.13 \pm 0.04$ | $0.04 \pm 0.01$ |

### D.4 Stopping criterion

In Algorithm 3, we update the removed feature from $Y$ with samples $X$. Suppose, we have observations $X_1, \ldots, X_N \sim P_X$, and $Y_1, \ldots, Y_N \sim P_Y$. Without any drifted components, we have $P_X = P_Y$ with

$$m = \mathbb{E}[X] = \mathbb{E}[Y] \in \mathbb{R}$$

$$\Sigma = \text{Cov}(X) = \text{Cov}(Y) \in S_+^d$$

where $m_X = \frac{1}{N} \sum_i^N X_i$, and $m_Y = \frac{1}{N} \sum_{i=1}^N Y_i$ denote the sample means and $S_+^d$ denotes the set of symmetric p.s.d. $d \times d$ matrices. We consider

$$||D|| = ||m(X) - m(Y)||,$$

then

$$\mathbb{E}[||D||] \leq \sqrt{\frac{2}{N} tr(\Sigma)}$$

Since we have $D \sim \mathcal{N}(0, \frac{2}{N}\Sigma)$, we can decompose $\Sigma = U\Lambda U^T$. Then with $Z = U^T D$, it follows $Z \sim \mathcal{N}(0, \frac{2}{N}\Lambda)$. Thus $||D||^2 = \sum_{i=1}^{d} \frac{2}{N}\lambda_i \chi_1^2$, with $\chi_1^2$ denotes a chi-squared distribution with one degree of freedom. Note that $\mathrm{tr}(\Sigma) = \sum_{i=1}^{d} \lambda_i$, where $\lambda_i$ is the $i$-th eigenvalue for $i = 1, \ldots, d$. Therefore, we have

$$\mathbb{E}||D||^2 = \frac{2}{N} tr(\Sigma),$$

applying Jensen inequality yields

$$\mathbb{E}[||D||] \leq \sqrt{\frac{2}{N} tr(\Sigma)}.$$

### D.5 Sensitivity Analysis

In the following, we investigate the sensitivity of Algorithm 3. Shows the sensitivity of explanations from Algorithm 3 to changes in the number of dimensions $d$, the number of samples $N$, the number of random projections $L$, and the quantile-level $q$.

Default parameters are $N = 500, L = 1000, q = 0.95, d = 50$, the underlying stream consists of two Mixture distributions (Gaussian/Exponential 50/50) with means randomly sampled in $(-3, 3)$ and variance $I_d$. We randomly select 5/10/15 components for which the mean is randomly changed by an offset uniformly sampled in $(-1, 1)$. In Figure 17, we visualize the norm of the mean differences and the removed components and the iterative procedure of Algorithm 2 for 20 removals in the proposed Algorithm 3. All components in the gray area would not be selected under the proposed stopping criteria. All results are averaged over five different runs with fixed random seeds. The red line indicates the ground truth features which exhibit a drift starting with 5 (left), 10 (middle), and 15 (right) for each subplot respectively.

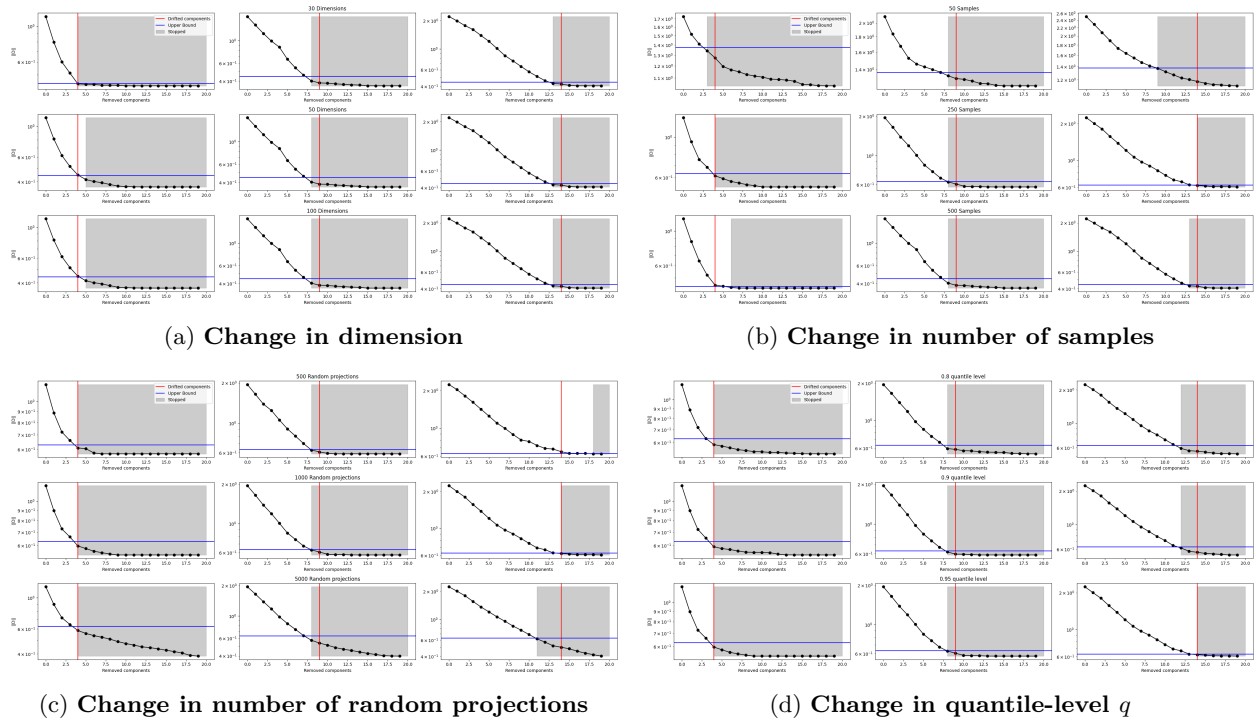

(a) **Change in dimension**  (b) **Change in number of samples**

(c) **Change in number of random projections**  (d) **Change in quantile-level** $q$

Figure 17: Sensitivity analysis of Algorithm 3 for a change in (a) dimension, (b) number of samples, (c) number of random projections, and (d) quantile-level.

# E Omitted Proofs

**Theorem 3.1.** *(Asymptotic law of $S_d(\theta)$) Assume (A1)–(A4). Let $\delta := \mu_P - \mu_Q$. Define the random vector*

$$U_d(\theta) := \begin{pmatrix} u_{1,d}(\theta) \\ u_{2,d}(\theta) \end{pmatrix} := \begin{pmatrix} \theta^\top \delta \\ \sqrt{\theta^\top \Sigma_P \theta} - \sqrt{\theta^\top \Sigma_Q \theta} \end{pmatrix},$$

*and let $\sqrt{d}\,\Delta_d \to m_2$, where $\Delta_d = \sqrt{\frac{1}{d}\operatorname{tr}(\Sigma_P)} - \sqrt{\frac{1}{d}\operatorname{tr}(\Sigma_Q)}$, with $m_2 \in \mathbb{R}$. Then, under (A2), the population slice statistic satisfies*

$$S_d(\theta) = W_2^2(P_\theta, Q_\theta) = \big(u_{1,d}(\theta)\big)^2 + \big(u_{2,d}(\theta)\big)^2 + r_d(\theta), \tag{2}$$

*where $r_d(\theta) \to 0$ in probability as $d \to \infty$ (the error stems only from the projection-to-Gaussian approximation in (A2)). Moreover, as $d \to \infty$, there exist centering/scaling constants such that*

$$\begin{pmatrix} \sqrt{d}\,u_{1,d}(\theta) \\ \sqrt{d}\,u_{2,d}(\theta) \end{pmatrix} \Rightarrow \mathcal{N}\left(\begin{pmatrix} 0 \\ m_2 \end{pmatrix},\, \Omega\right),$$

*some $2 \times 2$ covariance matrix $\Omega \succeq 0$. Consequently, $d\,S_d(\theta)$ converges in distribution to a (possibly noncentral) generalized chi-square random variable, i.e.*

$$d\,S_d(\theta) \Rightarrow Z^\top A Z,$$

*where $Z \sim \mathcal{N}(\mu_Z, \Sigma_Z)$ is Gaussian and $A \succeq 0$ is a fixed matrix. In particular, the limiting law is supported on $\mathbb{R}_+$.*

*Proof.* Step 1 (Gaussian closed form). Under (A2), for large $d$ the projected laws are well-approximated by Gaussians $\mathcal{N}(m_P(\theta), v_P(\theta))$ and $\mathcal{N}(m_Q(\theta), v_Q(\theta))$. For one-dimensional Gaussians, the squared 2-Wasserstein distance has the exact closed form

$$W_2^2\big(\mathcal{N}(m_1, s_1^2), \mathcal{N}(m_2, s_2^2)\big) = (m_1 - m_2)^2 + (s_1 - s_2)^2.$$

Plugging $m_1 = m_P(\theta)$, $m_2 = m_Q(\theta)$, $s_1 = \sqrt{v_P(\theta)}$, $s_2 = \sqrt{v_Q(\theta)}$ yields equation 2 with an error term $r_d(\theta)$ that vanishes in probability as $d \to \infty$ by the assumed projection-to-Gaussian approximation (A2). This proves equation 2.

Step 2 (Asymptotics of the linear term). Let $\delta = \mu_P - \mu_Q$. For $\theta \sim \operatorname{Unif}(\mathbb{S}^{d-1})$, one has $\mathbb{E}[\theta] = 0$ and $\operatorname{Cov}(\theta) = \frac{1}{d}I_d$, hence $\mathbb{E}[\theta^\top \delta] = 0$ and $\operatorname{Var}(\theta^\top \delta) = \|\delta\|_2^2/d$. Under standard spherical CLT conditions (covered by (A4)), $\sqrt{d}\,\theta^\top \delta \Rightarrow \mathcal{N}(0, \|\delta\|_2^2)$.

Step 3 (Asymptotics of the quadratic terms and delta method). Let $v_P(\theta) = \theta^\top \Sigma_P \theta$ and $v_Q(\theta) = \theta^\top \Sigma_Q \theta$. Under (A3)–(A4), the centered/scaled quadratic forms are jointly asymptotically normal:

$$\sqrt{d}\left(\begin{pmatrix} v_P(\theta) \\ v_Q(\theta) \end{pmatrix} - \begin{pmatrix} \operatorname{tr}(\Sigma_P)/d \\ \operatorname{tr}(\Sigma_Q)/d \end{pmatrix}\right) \Rightarrow \mathcal{N}(0, \Xi)$$

for some finite $2 \times 2$ covariance matrix $\Xi$ (depending on the limiting traces in (A3) and the cross-trace structure in (A4)). Now consider the smooth map $h(a, b) = \sqrt{a} - \sqrt{b}$ on $(0, \infty)^2$. By the multivariate delta method applied at $(\tau_P, \tau_Q)$,

$$\sqrt{d}\left(\sqrt{v_P(\theta)} - \sqrt{v_Q(\theta)} - (\sqrt{\tau_P} - \sqrt{\tau_Q})\right) \Rightarrow \mathcal{N}(0, \nabla h^\top \Xi \nabla h),$$

where $\nabla h(\tau_P, \tau_Q) = \left(\frac{1}{2\sqrt{\tau_P}}, -\frac{1}{2\sqrt{\tau_Q}}\right)^\top$. Thus $\sqrt{d}\,u_{2,d}(\theta)$ is asymptotically normal with mean $m_2 = \sqrt{d}(\sqrt{\tau_P} - \sqrt{\tau_Q})$ if $\tau_P \neq \tau_Q$ (and $m_2 = 0$ if $\tau_P = \tau_Q$), and finite variance given by the delta-method expression above.

Step 4 (Joint convergence and quadratic form). By (A4), the linear form $u_{1,d}(\theta)$ is asymptotically independent of the quadratic-form fluctuations that drive $u_{2,d}(\theta)$, hence the vector $(\sqrt{d}\,u_{1,d}(\theta), \sqrt{d}\,u_{2,d}(\theta))^\top$ converges jointly to a bivariate normal. Finally, equation 2 shows that

$$d\,S_d(\theta) = (\sqrt{d}\,u_{1,d}(\theta))^2 + (\sqrt{d}\,u_{2,d}(\theta))^2 + d\,r_d(\theta).$$

Since $d\,r_d(\theta) \to 0$ in probability (by (A2) and boundedness in (A3)), Slutsky's lemma yields that $d\,S_d(\theta)$ converges to a quadratic form in a Gaussian vector, i.e. a generalized chi-square random variable on $\mathbb{R}_+$. $\square$

**Proposition E.1.** *Suppose some i.i.d. samples $X_n = (x_1, \ldots, x_n)$ with $x_i \sim \Gamma(\alpha, \beta)$ for $i = 1, \ldots, n$ with sample mean $\overline{X}_n = \frac{1}{n}\sum_{i=1}^n x_i$ and sample variance $S_n^2 = \frac{1}{n-1}\sum_{i=1}^n (x_i - \overline{X}_n)^2$. Then, the two-tailed confidence intervals for confidence level $p$ of the Method of Moments (MoM) estimates $\widehat{\alpha}, \widehat{\beta}$ are*

$$\begin{aligned}
C_p(\widehat{\alpha}) &= \left[\widehat{\alpha} - z_{\frac{q}{2}} \cdot \sqrt{\mathrm{Var}(\widehat{\alpha})}, \widehat{\alpha} + z_{\frac{q}{2}} \cdot \sqrt{\mathrm{Var}(\widehat{\alpha})}\right] \\
C_p(\widehat{\beta}) &= \left[\widehat{\beta} - z_{\frac{q}{2}} \cdot \sqrt{\mathrm{Var}(\widehat{\beta})}, \widehat{\beta} + z_{\frac{q}{2}} \cdot \sqrt{\mathrm{Var}(\widehat{\beta})}\right]
\end{aligned} \tag{5}$$

*where $z_{\frac{q}{2}}$ is the z-value of a standard normal distribution for confidence level $q$, and*

$$\mathrm{Var}(\hat{\alpha}) \approx \frac{6\alpha^2}{n}, \quad \mathrm{Var}(\hat{\beta}) \approx \frac{\beta^2 + 2\alpha\beta^2}{n\alpha}$$

*Proof.* Suppose, we have i.i.d. samples $x_1, \ldots, x_n \sim \Gamma(\alpha, \beta)$ which we denote as $X_n$. For a Gamma distribution with shape $\alpha$ and rate $\beta$, we have $\mu = \frac{\alpha}{\beta}$ and $\sigma^2 = \frac{\alpha}{\beta^2}$. We write $\overline{X}_n = \frac{1}{n}\sum_{i=1}^n x_i$ for the sample mean and $S_n^2 = \frac{1}{n-1}\sum_{i=1}^n (x_i - \overline{X}_n)^2$ for the sample variance. Then, we have the following Method of Moment estimates for $\alpha$ and $\beta$

$$\widehat{\alpha} = \frac{\overline{X}_n^2}{S_n^2}, \quad \widehat{\beta} = \frac{\overline{X}_n}{S_n^2}.$$

By the Central Limit Theorem, we know that for large $n$, the sample mean and variance converges to a normal distribution, with

$$\sqrt{n}\left(\widehat{\alpha}\widehat{\beta}^{-1} - \mu\right) \xrightarrow{d} \mathcal{N}\left(0, \sigma^2\right)$$

$$\sqrt{n}\left(S_n^2 - \sigma^2\right) \xrightarrow{d} \mathcal{N}\left(0, \mathrm{Var}(S_n^2)\right)$$

where, with *Theorem 1* from Cho & Cho (2008), $\mathrm{Var}(S_n^2) \approx n^{-1}(3\sigma^2 + 2\sigma^2\mu2 - \sigma^4) = \frac{2\alpha^2}{n\beta^4}$ for $n \to \infty$. We use the asymptotic normality of sample mean and variance and apply the delta method to derive an approximation of the variance of $\hat{\alpha}, \hat{\beta}$. For a smooth differentiable function $g(\theta)$ and a sequence of random variables $\theta_n$, if $\sqrt{n}(\theta_n - \theta) \xrightarrow{d} \mathcal{N}(0, \Sigma)$, then $\sqrt{(n)}(g(\theta_n) - g(\theta)) \xrightarrow{d} \mathcal{N}\left(0, \nabla g(\theta)^T \Sigma \nabla g(\theta)\right)$. Beginning with the estimate for $\alpha$, we set

$$g(\overline{X}_n, S_n^2) = \frac{\overline{X}_n^2}{S_n^2},$$

with

$$\nabla g\left(\overline{X}_n^2, S_n^2\right)^T = \left(2\frac{\overline{X}_n}{S_n^2}, -\frac{\overline{X}_n^2}{(S_n^2)^2}\right).$$

The covariance matrix $\Sigma$ consists of $\mathrm{Var}(\overline{X}_n)$ and $\mathrm{Var}(S_n^2)$ on the diagonal and 0 on the off diagonal elements due to the fact that for large $n$ sample mean and variance are uncorrelated. Therefore, we have

$$\mathrm{Var}(\hat{\alpha}) \approx \left(\frac{2\overline{X}_n}{S_n^2}\right)^2 \cdot \mathrm{Var}(\overline{X}_n) + \left(\frac{\overline{X}_n^2}{(S_n^2)^2}\right)^2 \cdot \mathrm{Var}(S_n^2),$$

and plugging the estimator for sample mean and variance in, we may simplify the expression to

$$\text{Var}(\hat{\alpha}) \approx \frac{4\alpha^2}{n} + \beta^4 \cdot \text{Var}(S_n^2) = \frac{6\alpha^2}{n}.$$

For the estimator of $\beta$, we set

$$g(\overline{X}_n, S_n^2) = \frac{\overline{X}_n}{S_n^2},$$

repeating the steps from above leads to,

$$\text{Var}(\hat{\beta}) \approx \left(\frac{1}{S_n^2}\right)^2 \cdot \text{Var}(\overline{X}_n) + \left(\frac{\overline{X}_n}{(S_n^2)^2}\right)^2 \cdot \text{Var}(S_n^2),$$

which we simplify to

$$\text{Var}(\hat{\beta}) \approx \frac{\beta^2}{n \cdot \alpha} + \frac{\beta^6}{\alpha^2} \cdot \text{Var}(S_n^2).$$

$\square$

