# OpenReview forum: "High-Dimensional Online Change Point Detection with Adaptive Thresholding and Interpretability"
_TMLR — Decision pending for TMLR_

### Review · Reviewer_DSsZ · 2026-06-10

**Summary Of Contributions:**

This paper proposes SWCPD, an online change-point detection method for high-dimensional data streams. The main idea is to compare two adjacent time windows using the Sliced Wasserstein distance. Instead of computing a full high-dimensional optimal transport distance, which can be computationally expensive, the method repeatedly projects the data onto one-dimensional random directions, computes one-dimensional Wasserstein distances, and aggregates the resulting slice statistics. The paper argues that this makes the method more scalable for high-dimensional streams. The paper’s main technical contribution is an adaptive thresholding mechanism. The authors analyze the distribution of random Sliced Wasserstein slice statistics and motivate a Gamma approximation for these slice values. They then fit a Gamma distribution using method-of-moments estimates and use its quantiles to produce a dynamic threshold for online detection. The algorithm is presented as a CUSUM-style procedure applied to Sliced Wasserstein-based scores, where the novelty is mainly in the Gamma-motivated adaptive threshold and the explanation mechanism, rather than in cumulative monitoring itself.

The paper also proposes an interpretability module for detected change points. The explanation method identifies projection directions with large projected Wasserstein discrepancies, aggregates the absolute projection weights to produce feature-level importance scores, and then iteratively removes or masks important features to validate whether the apparent drift disappears. The authors describe these explanations as contrastive descriptions of distributional change rather than causal explanations.

Empirically, the paper evaluates SWCPD on synthetic data and several real-world datasets, including MNIST, HAR, HASC, and Occupancy. It compares against a broad range of online and offline change-point detection methods, including BOCPD, e-divisive, KCP, ClaSP, OT-CPD, RuLSIF, DeepRuLSIF, and DeepCLF. The reported results suggest that SWCPD often achieves low false-positive rates while maintaining competitive AUC, covering score, and detection delay.

**Audience:**

Yes

**Audience Explanation:**

The topic is relevant to machine learning researchers working on distribution shift, online monitoring, time-series segmentation, high-dimensional statistics, optimal transport, and interpretable ML. The practical motivation is strong: many deployed ML systems need to detect distributional changes in streaming data, and in high-dimensional settings it is often not enough to simply raise an alarm; users also need some explanation of what changed. The paper addresses this combination of scalability, online detection, and interpretability, which is likely to be interesting to TMLR readers.

The paper is also potentially interesting because it connects several areas that are usually treated separately: Sliced Wasserstein distances, online change-point detection, adaptive thresholding, and feature-level explanations. The use of random projections makes the approach relatively simple and computationally attractive, and the experiments suggest that this simplicity can work well in practice. Overall, I think the contribution of the paper is incremental but useful.

**Broader Impact Concerns:**

N/A.

**Claims And Evidence:**

Yes

**Claims Explanation:**

I think the claims are partially supported, some claims are on the stronger side.

The empirical claim that SWCPD can reduce false positives while remaining competitive is reasonably supported. In the offline comparison, SWCPD has low false positives across the real-world datasets and performs particularly well on HASC, where it reports strong AUC, very low false positives, and substantially lower detection delay than several baselines. In the online comparison, SWCPD also has strong AUC and low false positives on Occupancy, MNIST, HASC, and HAR, although it is not always best on covering score or detection delay. For example, on MNIST, DeepCLF and DeepRuLSIF have shorter detection delays than SWCPD, and on HAR, DeepRuLSIF and DeepCLF have better covering scores than SWCPD. Therefore, I think the empirical evidence supports a more precise claim: SWCPD offers a favorable false-positive/reliability trade-off, rather than being uniformly superior across all detection metrics.

My biggest concern is the theoretical justification for the Gamma threshold. The paper’s own theorem states that the general limiting distribution of the slice statistic is a generalized chi-square, while the Gamma limit appears only under a more specific mean-shift-dominated regime where the variance term is negligible. The paper then uses a Gamma fit as the operational thresholding mechanism more generally. This may be a useful empirical approximation, but the current theory does not fully justify the Gamma threshold for all types of drift considered in the experiments, especially changes involving variance or covariance. The paper should be clearer about when the Gamma approximation is theoretically justified and when it is mainly a modeling heuristic.

There is also a possible technical issue in the asymptotic argument. The theorem states that $\sqrt{d}u_{2,d}(\theta)$ converges with a finite mean $m_2$. But if the limiting average variances $\tau_P$ and $\tau_Q$ differ, then $\sqrt{\tau_P}-\sqrt{\tau_Q}$ is generally nonzero, so multiplying by $\sqrt{d}$ would diverge unless a local-alternative condition is imposed. This seems to conflict with the statement that the limiting mean is finite. The proof should either impose a local covariance-difference condition, assume $\tau_P=\tau_Q$, or revise the theorem. Since this theorem motivates the thresholding procedure, this issue is important.

**Requested Changes:**

1. Clarify and correct the theoretical justification for the Gamma threshold: The authors should clearly distinguish between three cases: the general asymptotic result, the special mean-shift-dominated Gamma case, and the empirical use of a Gamma approximation outside that special case. The current theorem says the general limit is a generalized chi-square distribution, while the Gamma limit only follows under additional assumptions. The paper should not imply that the Gamma threshold is theoretically guaranteed in general unless this can be proven. The authors should also revisit the asymptotic treatment of the variance term $u_{2,d}(\theta)$. As written, the finite limiting mean for $\sqrt{d}u_{2,d}(\theta)$ seems questionable when $\tau_P \neq \tau_Q$. This should be corrected by adding the necessary local-alternative assumption, restricting the theorem to $\tau_P=\tau_Q$, or rewriting the result.

2. Clarify the notation and algorithmic details around $q$: The notation around $q, 1-q$, and $\kappa(q)$ should be made fully consistent. In the text, the threshold is described as a $(1-q)$-quantile, while Algorithm 1 writes $U_t = C_t + \kappa_{t+1}(q)$. In Algorithm 2, $q$ is also used for selecting high-discrepancy slices for explanations. These may be related but should be clearly separated or renamed. Because $q$ directly affects the detector’s sensitivity and false positives, this ambiguity is not just cosmetic. The paper should define whether $q$ is a significance level, an upper-tail probability, or a quantile level, and use that definition consistently throughout.

3. Minor writing and presentation issues: The paper should be carefully proofread. I noticed inconsistent naming such as SWCPD/SWCDP, inconsistent notation for $q$, and minor typos such as “into into.” Some baseline names also appear inconsistently, for example RuLSIF/RuLIFS. These issues are not central, but they reduce clarity.

---

> ### Author Response · Authors · 2026-06-25
>
> We sincerely thank the reviewer for their careful reading and insightful feedback. Below, we comment on the requested changes.
>
> # Clarification Gamma threshold
> We have revised the theoretical section accordingly. In the general case, the limiting distribution of the scaled sliced statistic is a generalized chi-square distribution, not necessarily a Gamma distribution. A Gamma limit only follows under additional simplifying assumptions, for example, in a mean-shift-dominated setting where the variance contribution is asymptotically negligible, and the statistic reduces to a scaled chi-square term. Outside such special cases, the Gamma threshold should be understood as a moment-matched or empirical approximation to the null distribution and not as a theoretically guaranteed limit.
> We also correct the asymptotic treatment of the variance term $u_{2,d}(\theta)$. As the reviewer notes, if $\tau_P \neq \tau_Q$, then $\sqrt{d}u_{2,d}(\theta)$ does not have a finite limiting mean in general. We therefore revise the theorem to make the required condition explicit: either the variance term is considered under $\tau_P=\tau_Q$, or a local-alternative condition is imposed such that $\sqrt{d}(\sqrt{\tau_P}-\sqrt{\tau_Q})$ converges to a finite constant. Without such a condition, the scaled statistic may diverge instead of converging to a finite generalized chi-square distribution.
>
> Even when the full limit is a generalized chi-square (Theorem 3.1), the slice statistic is nonnegative and often well-approximated by a Gamma distribution in practice. This is precisely the modeling assumption used by SWCPD.
> We will make clear in the revised manuscript that the theoretical guarantee is the generalized chi-square limit under the stated assumptions, while the Gamma threshold is theoretically justified only in the corresponding special case and otherwise used as an empirical approximation.
>
> We thank the reviewer for pointing out editorial improvements, which we will incorporate in our revised manuscript.

---

### Review · Reviewer_p6Cj · 2026-06-19

**Summary Of Contributions:**

## Summary
This paper proposes a framework for high-dimensional online change point detection, aiming to improve both scalability and explainability. The method first reduces the multivariate sequential data to a one-dimensional representation via random projection, and then constructs the monitoring statistic using the sliced Wasserstein distance. It further introduces an adaptive quantile-based thresholding strategy for online change point detection, together with a contrastive explanation mechanism to enhance interpretability. Theoretical analysis is provided to support threshold calibration, and experiments indicate improved detection performance over competing baselines.

## Strengths
- The problem studied is important and well-motivated for real-world applications.
- The experimental evaluation appears to be comprehensive.
## Weaknesses and Questions
- The proposed method does not achieve sufficiently strong performance on the MNIST and HAR datasets. In terms of the metrics COV and DD, there is a noticeable gap compared with the best-performing baselines. The authors should clarify the underlying reasons for this discrepancy and discuss whether it is due to the assumptions, the parameter settings, or the characteristics of these datasets.
- The theoretical analysis relies on assumptions A1–A4, which are nontrivial and technically strong. In practical settings involving abrupt and nonstationary distributional shifts, such as financial time series or network-attack traffic, the data may exhibit heavy-tailed behavior. Would heavy tails violate the assumptions underlying the theoretical derivations? If so, how significantly could such violations affect both the theoretical guarantees and the empirical findings?
- Are the experimental details for the baseline methods kept consistent to ensure a fair comparison? In addition, how are hyperparameters for the baselines selected? Providing these details would improve reproducibility and fairness.
- There are multiple typos in the manuscript (e.g., “into into”, “neuronal networks”). The authors should carefully proofread and correct these language and formatting errors.

**Audience:**

Yes

**Audience Explanation:**

Researchers in the fields of time series analysis and machine learning would be interested in this paper.

**Claims And Evidence:**

Yes

**Claims Explanation:**

Partly agree. Further discussion of the theoretical analysis and the experimental details mentioned in the weakness section would strengthen the credibility of the paper.

**Requested Changes:**

Please see the weakness.

---

> ### Author Response · Authors · 2026-06-25
>
> We sincerely thank the reviewer for their careful reading and insightful feedback. Below, we comment on the requested changes.
> # Performance HAR & MNIST
> The larger DD values are mainly caused by the sliding-window design of SWCPD. A change can only be detected once enough post-change observations are available within the window.
> In contrast, the gap in COV is primarily due to the conservative behavior of SWCPD. The method is tuned to avoid false positives and therefore tends to predict fewer change points than more sensitive baselines. This can reduce spurious detections, but it may also miss true changes or annotate some sequences as unchanged. Since COV measures the overlap between predicted and ground-truth segmentations, missed changes can merge several true segments into one predicted segment and lower the covering score. We will clarify this trade-off between low false positives and segmentation coverage in the revised manuscript.
> # Heavy-tailed data
> A limitation of the proposed method is that the theoretical guarantees rely on Assumptions A1–A4. These assumptions impose some regularity conditions on the underlying distributions and may not hold for strongly heavy-tailed or highly nonstationary data. If such assumptions are violated, the concentration behavior used in the theoretical analysis may be violated, and the guarantees no longer apply as stated. In practice, this can lead to degraded performance, for example, through unstable distance estimates, increased sensitivity to extreme observations, or less reliable thresholding. We therefore include diagnostic checks of the assumptions in the appendix and view robust extensions for heavy-tailed settings as an important direction for future work.
> # Experimental Setup
> We agree that these details are important for reproducibility and fairness. All baseline methods were evaluated under the same experimental protocol, using identical datasets, preprocessing, and evaluation metrics. Their hyperparameters were manually tuned to obtain competitive performance for each baseline, and we provide a comprehensive overview of the selected values and implementation details in the appendix.
>
> We thank the reviewer for pointing out non-critical editorial improvements, which we will incorporate in our revised manuscript.

---

### Review · Reviewer_ttfw · 2026-06-20

**Summary Of Contributions:**

### Summary

The paper introduces an online change point detection algorithm sliced Wasserstein distance change point detection (SWCPD). SWCPD detects the change point by measuring the sliced Wasserstein distance between two consecutive subsequences. If the distance is greater than a threshold, the algorithm concludes that a change has occurred. The threshold is adaptively determined based on the theoretical analysis that the SW distance can be approximated by a Gamma distribution.

### Strengths

* The algorithm is based on a sound theoretical analysis on the distribution of SW distances.
* The algorithm shows competitive performance compared to existing CPD methods.

### Weaknesses

* The experimental results on the Occupancy dataset is questionable as the standard deviations are all zero.
* It is unclear why Table 1 compares the SWD explanations with IG, GS, and DL explanations instead of ground-truth features, which I believe would be accessible due to the synthetic nature of the data.

**Audience:**

Yes

**Audience Explanation:**

The paper presents a novel online CPD algorithm that can be used in multiple real-world applications.

**Broader Impact Concerns:**

I do not have any ethical concerns specific to this work, and I do not believe a Broader Impact Statement is strictly necessary, though I note that the paper does not include one.

**Claims And Evidence:**

No

**Claims Explanation:**

In Tables 4 and 5, the experimental results on the Occupancy dataset has zero standard deviation. Considering the inherent randomness of SWD, this seems highly unlikely.

**Requested Changes:**

### Critical

* It is unclear why Table 1 compares the SWD explanations with IG, GS, and DL explanations instead of ground-truth features, which I believe would be accessible due to the synthetic nature of the data. Also, the authors need to provide the strengths of SWCPD explanations compared to IG, GS, and DL explanations. Without them, there are no reason for one to adopt the SWCPD method instead of existing methods to generate explanations.
* The zero standard deviations for the Occupancy dataset results in Tables 4 and 5 are questionable. If this is just a result of rounding, it would be great if the authors could explicitly mention it in the main paper / caption and briefly explain why the standard deviations are so small for the Occupancy dataset.

### Non-Critical

* In Algorithm 1, the Project function is called on line 8, but its definition cannot be found.
* The paper states that sliding window is partitioned into two subsequences of length $\lfloor w/2\rfloor$ and $w-\lfloor w/2\rfloor$. Then, shouldn't the detected change point be $t+\lfloor w/2\rfloor$ instead of $t+w$ on line 14 of Alrgorithm 1?

---

> ### Author Response · Authors · 2026-06-25
>
> We sincerely thank the reviewer for their careful reading and insightful feedback. Below, we comment on the requested changes.
> # Table 1
> We thank the reviewer for suggesting adding a comparison to the ground truth features. To address the reviewer’s concern, we have revised and extended the experimental evaluation and now additionally compare all explanation methods directly against the ground-truth drift features. The revised Table 1, therefore, reports the agreement of IG, GS, DL, and SWCPD with the true drift features as reported below.
> |        | $d = 10$ IG    | $d = 10$ GS    | $d = 10$ DL    | $d = 10$ SWD   | $d = 20$ IG    | $d = 20$ GS    | $d = 20$ DL    | $d = 20$ SWD   |
> |:--------|:------------------|:------------------|:------------------|:------------------|:------------------|:------------------|:------------------|:------------------|
> | $k = 1$ | **0.991** $\pm$ **0.010** | $0.987 \pm 0.015$ | $0.982 \pm 0.021$ | $0.961 \pm 0.057$ | $0.999 \pm 0.000$ | $0.999 \pm 0.000$ | $0.998 \pm 0.002$ | $0.991 \pm 0.003$ |
> | $k = 3$ | $0.928 \pm 0.052$ | $0.929 \pm 0.051$ | $0.930 \pm 0.052$ | **0.997 $\pm$ 0.002** | $0.958 \pm 0.020$ | $0.957 \pm 0.021$ | $0.959 \pm 0.016$ | **0.991 $\pm$ 0.003** |
> | $k = 7$ | $0.915 \pm 0.041$ | $0.916 \pm 0.040$ | $0.911 \pm 0.040$ | **0.997 $\pm$ 0.001** | $0.933 \pm 0.007$ | $0.933 \pm 0.007$ | $0.934 \pm 0.016$ | **0.994 $\pm$ 0.003** |
> | $k = 9$ | $0.893 \pm 0.040$ | $0.894 \pm 0.040$ | $0.898 \pm 0.034$ | **0.998 $\pm$ 0.000** | $0.913 \pm 0.020$ | $0.913 \pm 0.020$ | $0.908 \pm 0.031$ | **0.994 $\pm$ 0.001** |
>
> # Zero Standard Deviation
> We thank the reviewer for pointing this out. The zero standard deviations for the Occupancy results are not due to rounding and should indeed have been clarified. In our experimental setup, the reported standard deviations are computed across datasets within each benchmark dataset. Since Occupancy consists of only a single dataset in our evaluation, there is no within-dataset variability to estimate for this case. The displayed value, therefore, reflects the absence of multiple datasets and not an empirical standard deviation of zero.
> We will clarify this in the main paper and in the captions of Tables 4 and 5, and replace/annotate the corresponding entries to avoid suggesting that repeated runs produced exactly zero variance.
>
> We thank the reviewer for pointing out non-critical editorial improvements, which we will incorporate in our revised manuscript.

---

> > ### Comment · Reviewer_ttfw · 2026-07-02
> >
> > Thank you for the clarification. While the table clearly shows that SWD features align better with the ground truth than the IG, GS, and DL features, it would be helpful to know whether SWD offers any other qualitative advantages over these methods.
> > Additionally, as I understand it, SWD depends on the selection of $\theta$, which is sampled from $\text{Unif}(\mathbb{S}^{d-1})$. Doesn't this imply that SWCPD is also inherently random, producing different results across seeds even though there is only one dataset?

---

> > > ### Author Response · Authors · 2026-07-02
> > >
> > > We thank you for your engagement and follow-up comment.
> > >
> > > Within the scope of this work, which focuses on interpretable high-dimensional change point detection, SWD provides qualitative advantages beyond quantitative alignment scores. In particular, SWD provides detector-native contrastive explanations, identifying which features differ most between the reference and current distribution using the same distributional discrepancy that drives detection. In contrast, IG, GS, and DL require a separately trained predictive model and explain that model’s attribution differences rather than the change-point statistic directly.
> > >
> > > We thank the reviewer for raising this point. SWCPD is indeed stochastic in its finite-sample implementation because the Sliced Wasserstein distance is approximated using randomly sampled projection directions $\theta \sim \mathrm{Unif}(\mathbb{S}^{d-1})$. The randomness only enters through the Monte Carlo approximation. As the number of projections $L$ increases, this approximation becomes more stable (see Table 13, Figure 16). In the Occupancy experiment, the dimensionality is small $d=4$ and we use $L=1000$ projections, so the number of projections is much larger than the ambient dimension. To verify that the reported results are not driven by a particular random seed, we repeated the Occupancy experiment over ten (10) different seeds and observed no significant standard deviation in the evaluation metrics. We will add this clarification to the revised manuscript.
> > >
> > > | Metric |  SWCPD     |
> > > | ------ | --------------: |
> > > | AUC    | 0.5858 ± 0.0084 |
> > > | COV    | 0.8062 ± 0.0009 |
> > > | FP     | 4.0000 (4;4) |
> > > | DD     | 51.87 (49.54;52.8) |